# AOBLMOA: A Hybrid Biomimetic Optimization Algorithm for Numerical Optimization and Engineering Design Problems

**DOI:** 10.3390/biomimetics8040381

**Published:** 2023-08-21

**Authors:** Yanpu Zhao, Changsheng Huang, Mengjie Zhang, Yang Cui

**Affiliations:** 1School of Economics and Management, China University of Petroleum (East China), Qingdao 266580, China; z21080319@s.upc.edu.cn (Y.Z.); s21080054@s.upc.edu.cn (Y.C.); 2Dareway Software Co., Ltd., Jinan 250200, China; zmj@dareway.com.cn

**Keywords:** mayfly optimization algorithm, aquila optimizer, opposition-based learning, numerical optimization problems, engineering design problems, global optimization

## Abstract

The Mayfly Optimization Algorithm (MOA), as a new biomimetic metaheuristic algorithm with superior algorithm framework and optimization methods, plays a remarkable role in solving optimization problems. However, there are still shortcomings of convergence speed and local optimization in this algorithm. This paper proposes a metaheuristic algorithm for continuous and constrained global optimization problems, which combines the MOA, the Aquila Optimizer (AO), and the opposition-based learning (OBL) strategy, called AOBLMOA, to overcome the shortcomings of the MOA. The proposed algorithm first fuses the high soar with vertical stoop method and the low flight with slow descent attack method in the AO into the position movement process of the male mayfly population in the MOA. Then, it incorporates the contour flight with short glide attack and the walk and grab prey methods in the AO into the positional movement of female mayfly populations in the MOA. Finally, it replaces the gene mutation behavior of offspring mayfly populations in the MOA with the OBL strategy. To verify the optimization ability of the new algorithm, we conduct three sets of experiments. In the first experiment, we apply AOBLMOA to 19 benchmark functions to test whether it is the optimal strategy among multiple combined strategies. In the second experiment, we test AOBLMOA by using 30 CEC2017 numerical optimization problems and compare it with state-of-the-art metaheuristic algorithms. In the third experiment, 10 CEC2020 real-world constrained optimization problems are used to demonstrate the applicability of AOBLMOA to engineering design problems. The experimental results show that the proposed AOBLMOA is effective and superior and is feasible in numerical optimization problems and engineering design problems.

## 1. Introduction

There are a large number of optimization problems in the fields of mathematics and engineering, and these problems must be solved in a short time with highly complex constraints. However, some studies showed that such traditional methods fall into local optima when solving high-dimensional, multimodal, nondifferentiable, and discontinuous problems, resulting in low solution efficiency [1]. Compared with traditional methods, the metaheuristic algorithm has the characteristics of fewer parameters and no gradient information, so it can usually perform very well on such problems [2].

The Mayfly Optimization Algorithm (MOA) [3], proposed by Konstantinos Zervoudakis and Stelios Tsafarakis in 2020, is a metaheuristic algorithm inspired by the mating behavior of mayfly populations. Since it combines the main advantages of the particle swarm algorithm, genetic algorithm, and firefly algorithm, many researchers improved it and applied it to various scenarios. For example, Zhao et al. [4] presented a chaos-based mayfly algorithm with opposition-based learning and Lévy flight for numerical and engineering design problems. Zhou et al. [5] used orthogonal learning as well as chaotic strategies to improve the diversity of the MOA. Li et al. [6] proposed an IMA with chaotic initialization, the gravity coefficient, and a mutation strategy and applied it to the dynamic economic environment scheduling problem. Zhang et al. [7] combined the Sparrow Search Algorithm (SSA) with the MOA and applied it to the RFID network planning problem. Muhammad et al. [8] combined the MPA with the MOA and applied it to the maximum power tracking scenario in photovoltaic systems.

Based on the above, it can be found that using some strategies to optimize the MOA or combining other metaheuristic algorithms with the MOA can effectively improve the performance of the algorithm. However, although the above variants of the MOA improve the optimization ability of the MOA to varying degrees, they may present challenges when solving some non-convex optimization problems. Therefore, it is still necessary to find additional algorithms or methods to combine with the MOA to optimize it.

The Aquila Optimizer (AO), proposed by Abualigah et al. [9] in 2021, is a novel SIA that simulates the predation process of Aquila. Due to the variety of search methods for the AO, it received extensive attention from researchers. Mahajan et al. [10] combined the Arithmetic Optimization Algorithm (AOA) with the AO and applied it to the global optimization task. Ma et al. [11] combined GWO with the AO to construct a new hybrid algorithm. Ekinci et al. [12] proposed an enhanced AO as an effective control design method for automatic voltage regulators. Liu et al. [13] combined the WOA with the AO and applied it to the Cox proportional hazards model. From these studies, we find that the AO, as a metaheuristic algorithm that combines multiple optimization strategies, is mostly used by researchers in combination with other algorithms to optimize the performance of the other algorithms.

The opposition-based learning (OBL) strategy was proposed by Tizhoosh et al. [14]. Since the solution after OBL has a higher probability of being closer to the global optimal solution than the solution without OBL [15], it is often used by researchers to optimize the metaheuristic algorithm. For example, Zeng et al. [16] optimized the wild horse optimizer algorithm with OBL and applied the algorithm to the HWSN coverage problem. Muhammad et al. [17] combined OBL with teaching–learning-based optimization. Jia et al. [18] hybridized OBL with the Reptile Search Algorithm. Sarada et al. [19] mixed OBL with the Golden Jackal Optimization algorithm to optimize it. In these studies, OBL turned out to be effective at optimizing algorithms.

The no free lunch (NFL) [20] theorem states that no one optimization method can solve all optimization problems, and the MOA was shown to degrade the quality of the final solution owing to premature convergence [21]. Therefore, to enhance the global optimization ability of the MOA, we propose an MOA that combines the AO and the opposition-based learning (OBL) strategy, namely, AOBLMOA. The proposed algorithm integrates the search strategies in the AO into the position update process of male and female mayflies in the MOA, without increasing the time complexity, and adopts the OBL strategy for the offspring mayfly population to further optimize the offspring population. We also apply the hybrid algorithm to benchmark functions, CEC2017 numerical optimization problems, and CEC2020 real-world constrained optimization problems to show the feasibility of the proposed hybrid algorithm in optimization problems.

The organizational structure of this paper is as follows: Section 2 briefly describes the basic MOA, the AO, OBL, and other popular metaheuristic algorithms; Section 3 introduces the specific content of the proposed hybrid algorithm in detail; Section 4 analyzes the performance of AOBLMOA on benchmark functions, CEC2017 numerical optimization problems, and CEC2020 real-world constrained optimization problems; Section 5 summarizes the full text and looks forward to future work. The MATLAB codes of AOBLMOA are available at https://github.com/SheldonYnuup/AOBLMOA (accessed on 1 August 2023).

## 2. Background Overview

### 2.1. Popular Metaheuristic Algorithms

Metaheuristic algorithms (MHs) can generally be divided into four categories: Swarm Intelligence Algorithms (SIAs), Evolutionary Algorithms (EAs), Physics-based Algorithms (PhAs), and Human-based Algorithms (HAs).

SIAs mainly simulate various group behaviors of diverse organisms, including foraging, attacking, mating, and other behaviors. Such algorithms include Particle Swarm Optimization (PSO) [22], the Salp Search Algorithm (SSA) [23], Gray Wolf Optimization (GWO) [24], the Monarch Butterfly Optimizer (MBO) [25], the Bald Eagle Search Optimization Algorithm (BES) [26], the Marine Predator Algorithm (MPA) [27], the Whale Optimization Algorithm (WOA) [28], the Moth Search Algorithm (MS) [29], etc.

EAs mainly simulate the evolutionary process in nature, and the most classic algorithm is the Genetic Algorithm (GA) [30]. In recent years, with the in-depth study of EAs, researchers have successively proposed Differential Evolution (DE) [31], Biogeography-Based Optimization (BBO) [32], Cuckoo Search (CS) [33], and so on.

PhAs are mainly inspired by the laws of physics, including Multi-Verse Optimization (MVO) [34], Equilibrium Optimization (EO) [35], the Gravity Search Algorithm (GSA) [36], the Lightning Search Algorithm (LSA) [37], the Archimedes Optimization Algorithm (AOA) [38], and so on.

HAs simulate human social behavior. Existing HAs include teaching–earning-based optimization (TLBO) [39], the Human Felicity Algorithm (HFA) [40], Political Optimization (PO) [41], etc. Table 1 provides an overview of the mentioned metaheuristic algorithms.

### 2.2. Mayfly Optimization Algorithm (MOA)

The working principle of the mayfly algorithm is as follows: at the initial stage of the algorithm, male and female mayfly populations are randomly generated in the problem space. Each individual mayfly in the two populations represents a candidate solution in the problem space, which can be expressed by the d-dimensional variable $x=(x_{1},\cdot \cdot \cdot ,x_{d})$. Additionally, the performance of each individual mayfly is calculated by the objective function f(x). The velocity of each individual ephemera is represented by $v=(v_{1},\cdot \cdot \cdot ,v_{d})$, which is used to represent the change in position of each individual mayfly in each iteration. The movement direction of each individual mayfly is affected by both the individual optimal position pbest and the global optimal position gbest. That is, each individual mayfly adjusts its flight path to adapt to the individual optimal position pbest and the global optimal position gbest before the current iteration.

#### 2.2.1. Movement of Male Mayflies

Male mayfly groups tend to gather in the center of the group in space, which means that each individual male adjusts its position based on its own experience and the experience of its neighbors. $x_{ij}^{t}$ represents the position of individual i in dimension j at the t iteration, and $v_{ij}^{t}$ represents the velocity of individual i in dimension j at the t iteration. The movement of the male mayfly individual’s position $x_{ij}^{t}$ is shown in Equation (1):(1)$$x_{ij}^{t+1}=x_{ij}^{t}+v_{ij}^{t+1},$$

When the fitness of male mayflies is worse than the global optimal fitness, then male mayflies approach the individual optimal position and the global optimal position. In contrast, if the individual fitness of male mayflies is better than the global optimal fitness, then male mayflies perform courtship dance behaviors above the water surface to attract mates. Based on this, it should be assumed that although the male mayflies can continue to move, they cannot quickly gain speed. Taking the minimization problem as an example, the variation in the speed $v_{ij}^{t}$ of the male mayfly is shown in Equation (2):(2)$$v_{ij}^{t+1}=\left\{\begin{array}{c} {g*v}_{ij}^{t}+a_{1}e^{-\beta r_{p}^{2}}\left({pbest}_{ij}-x_{ij}^{t}\right)+\\ a_{2}e^{-\beta r_{g}^{2}}\left({gbest}_{j}-x_{ij}^{t}\right),f\left({gbest}_{j}\right)>f\left(x_{i}\right)\\ v_{ij}^{t}+d*r,f\left({gbest}_{j}\right)\leq f\left(x_{i}\right) \end{array}\right.,$$
where $f:R^{n}\to R$ represents the objective function, which is used to evaluate the performance of the solution; $v_{ij}^{t}$ represents the velocity of individual i in dimension j at the t-th iteration; $x_{ij}^{t}$ represents the position of individual i in dimension j at the t-th iteration; $a_{1}$ and $a_{2}$ represent the individual optimal attraction coefficient and the population optimal attraction coefficient, respectively; β represents the visibility coefficient, which is used to limit the visible range of individuals; d is the mating dance coefficient, which is used to attract the opposite sex to keep approaching; r is a random coefficient, which is valued in the range of [−1, 1]; and g represents the gravity coefficient, which is used to maintain the velocity of the individual in the previous iteration, and its expression is shown as
(3)$$g=g_{max}-\frac{g_{max}-g_{min}}{{iter}_{max}}\times iter,$$
where $g_{max}$ and $g_{min}$ represent the maximum and minimum values of the gravity coefficient, respectively; ${iter}_{max}$ indicates the maximum number of iterations of the algorithm, and iter indicates the current number of iterations of the algorithm.

$r_{p}$ and $r_{g}$ in Equation (2) represent the Cartesian distance between $x_{ij}^{t}$ and ${pbest}_{ij}$ and ${gbest}_{j}$, respectively. The Cartesian distance calculation formula is shown in Equation (4):(4)$$\left\|x_{i}-X_{i}\right\|=\sqrt{\sum_{j=1}^{n}{\left(x_{ij}-X_{ij}\right)}^{2}},$$
where $x_{ij}$ represents the individual position of male mayfly individual i in dimension j; $X_{ij}$ indicates the position of ${pbest}_{ij}$ or ${gbest}_{j}$; and ${pbest}_{ij}$ is the individual optimal position of mayfly individual i in dimension j. Taking the minimization problem as an example, the updated formula of the individual optimal position at the t+1 iteration is shown in Equation (5):(5)$${pbest}_{ij}=\left\{\begin{array}{c} x_{ij}^{t+1},f\left(x_{ij}^{t+1}\right)<f\left({pbest}_{ij}\right)\\ {pbest}_{ij},f\left(x_{ij}^{t+1}\right)\geq f\left({pbest}_{ij}\right) \end{array}\right.,$$
where ${gbest}_{j}$ is the optimal position of the group in the j dimension, and its updated formula is shown in Equation (6):(6)$$gbest\in \left\{{pbest}_{1},{pbest}_{2},\cdot \cdot \cdot ,{pbest}_{N}|f(cbest)\right\}=\min\left\{f\left({pbest}_{1}\right),f\left({pbest}_{2}\right),\cdot \cdot \cdot ,f\left({pbest}_{N}\right)\right\},$$
where N represents the number of individuals in the male mayfly population.

#### 2.2.2. Movement of Female Mayflies

The most striking difference between female mayflies and male mayflies is that females do not congregate in large groups. Instead, they mate by flying to males. Suppose $y_{ij}^{t}$ represents the position of individual i in dimension j at iteration t, and the position update formula of the female mayfly at iteration t+1 is shown in Equation (7):(7)$$y_{ij}^{t+1}=y_{ij}^{t}+v_{ij}^{t+1},$$

Although the process of the mutual attraction of mayflies is stochastic, this stochastic process can also be modeled as a deterministic process, that is, according to the mayflies’ fitness. For example, the female mayfly with the best fitness should be attracted by the male mayfly with the best fitness, and the female mayfly with the second-best fitness should be attracted by the male mayfly with the second-best fitness. By analogy, when the fitness of the female mayfly is inferior to that of the corresponding male mayfly, the female mayfly is attracted by the corresponding male mayfly and approaches it; otherwise, the female mayfly randomly moves. Therefore, taking the minimization problem as an example, the velocity update formula of the female mayfly is shown in Equation (8):(8)$$v_{ij}^{t+1}=\left\{\begin{array}{c} {g*v}_{ij}^{t}+a_{3}e^{-\beta r_{mf}^{2}}\left(x_{ij}^{t}-y_{ij}^{t}\right),f\left(y_{i}\right)>f\left(x_{i}\right)\\ {g*v}_{ij}^{t}+fl*r,f\left(y_{i}\right)\leq f\left(x_{i}\right) \end{array}\right.,$$
where $v_{ij}^{t}$ represents the velocity of individual i in dimension j at the t-th iteration. $a_{3}$ is the attraction coefficient between the male and female mayflies. β represents the visibility coefficient. $r_{mf}$ represents the Cartesian distance between the female mayfly and the male mayfly calculated by Equation (3). fl represents the random walk coefficient. r represents a random coefficient, taking values in the range [−1, 1].

#### 2.2.3. Mating of Male and Female Mayflies

The mating process of the two sexes of mayflies is represented by a crossover operator, and the survival of the fittest mechanism is used to select a male parent from the male mayfly population and a female parent and the female mayfly population for mating. That is, the male mayfly with the best fitness mates with the female mayfly with the best fitness, the male mayfly with the second-best fitness mates with the female mayfly with the second-best fitness, and so on. Before the end of each iteration, the male mayfly population, the female mayfly population, and the offspring mayfly population are merged into two populations, and then the individuals with the poorest fitness in the two merged populations is eliminated. The individuals with better fitness in the two merged populations enter the next iteration as the new male mayfly population and female mayfly population, respectively. The expressions of the offspring produced after mating are shown in Equations (9) and (10):(9)$$offspring1=L\times male+\left(1-L\right)*female,$$
(10)$$offspring2=L\times female+\left(1-L\right)*male,$$
where male and female represent the male parent and female parent, respectively; L represents a random value that conforms to a specific range; and the initial velocity of the offspring is set to zero.

#### 2.2.4. Mutation of Offspring Mayflies

To solve the problem that the algorithm may fall into a local optimum, the mutation operation is performed on the offspring mayflies, and, by adding random numbers that obey the normal distribution, the offspring mayflies can explore new areas that may not have been explored in the problem space. Among them, the number of mutant individuals is approximately 0.05 times that of the male mayflies. The expression of the offspring gene mutation is shown in Equation (11):(11)$${offspring}_{n}={offspring}_{n}+\sigma N_{n}\left(0,1\right),$$
where σ represents the standard deviation of the normal distribution. $N_{n}\left(0,1\right)$ represents a standard normal distribution with a mean of 0 and a variance of 1.

### 2.3. Aquila Optimizer (AO)

The AO simulates Aquila’s hunting behavior and optimizes the problem through its various methods in the hunting process. These methods include the high soar with vertical stoop, contour flight with short glide attack, low flight with slow descent attack, and diving to walk and grab prey. At the same time, the AO decides whether to use the exploration method or the exploitation method by judging the number of iterations. When the current iteration number is within 2/3 times the total iteration number, the explore method is fired; otherwise, the exploit method is used. The specific mathematical model of the method adopted by the AO is as follows:

#### 2.3.1. High Soar with Vertical Stoop (Expanded Exploration)

In this method, Aquila first identifies the location of the prey and selects the best prey area by the high soar with vertical stoop method. In this process, Aquila swoops from various locations within the problem space to ensure a wide area for the search space. The method can be represented as
(12)$$x_{ij}^{t+1}={gbest}_{j}\times \left(1-\frac{t}{T_{max}}\right)+\left(x_{M}^{t}-{gbest}_{j}\times \mathrm{rand}\right),$$
where $\left(1-\frac{t}{T_{max}}\right)$ is used to control the expanded search process by judging the number of iterations. t represents the current iteration number. $T_{max}$ represents the maximum number of iterations. rand represents a random value in the interval (0, 1). $x_{M}^{t}$ represents the average value of the individual x position as of the t-th iteration, which can be expressed as
(13)$$x_{M}^{t}=\frac{1}{N}\sum_{i=1}^{N}\;x_{ij}\left(t\right),\forall j=1,2,\ldots ,dim,$$
where N represents the population size, and dim represents the dimension of the problem to be solved.

#### 2.3.2. Contour Flight with Short Glide Attack (Narrowed Exploration)

The contour flight with short glide attack method mainly simulates the behavior of Aquila hovering above the target prey, preparing to land, and attacking after finding the target prey at high altitude. In this method, the algorithm more precisely explores the area where the target prey is located and prepares for the next attack. The simulation method can be mathematically expressed as
(14)$$x_{ij}^{t+1}={gbest}_{j}\times \mathrm{Levy}\left(D\right)+x_{R}^{t}+\left(f_{1}-f_{2}\right)\times rand,$$
where $x_{R}^{t}$ represents a random solution obtained in the population at the t-th iteration, D represents the problem dimension, $\mathrm{Levy}\left(D\right)$ is the Lévy flight distribution function, and its calculation formula can be shown as
(15)$$\mathrm{Levy}\left(D\right)=s\times \frac{u\times \sigma }{|v{|}^{\frac{1}{\beta }}},$$
where s is a constant value with a value of 0.01, u and v are both random numbers with a value in the (0, 1) interval, β is a constant value with a value of 1.5, and the calculation formula of σ is
(16)$$\sigma =\left(\frac{\Gamma \left(1+\beta \right)\times \sin e\left(\frac{\pi \beta }{2}\right)}{\Gamma \left(\frac{1+\beta }{2}\right)\times \beta \times 2\left(\frac{\beta -1}{2}\right)}\right),$$

y and x in Equation (14) are used to represent the spiral search method in the search process, and its calculation formula can be shown as
(17)$$f_{1}=r\times \cos\left(\theta \right),$$
(18)$$f_{2}=r\times \sin\left(\theta \right),$$
where
(19)$$r=r_{1}+U\times D_{1},$$
(20)$$\theta =-\omega \times D_{1}+\theta_{1},$$
(21)$$\theta_{1}=\frac{3\times \pi }{2},$$
where $r_{1}$ takes a value according to the number of search cycles in the interval [1, 20], U is a constant value with a value of 0.00565, $D_{1}$ is an integer value determined according to the dimension of the problem search space, and ω is a constant with a value of 0.005.

#### 2.3.3. Low Flight with Slow Descent Attack (Expanded Exploitation)

When the location of the prey is locked, Aquila needs to be ready to land and attack the prey. Therefore, Aquila tests the reaction of the prey by making an initial dive. In this method, the algorithm makes Aquila approach the prey and attack by exploiting the area where the target is located, which can be mathematically expressed as
(22)$$x_{ij}^{t+1}=\left({gbest}_{j}-x_{M}^{t}\right)\times \alpha -\mathrm{rand}+\left(\left(UB-LB\right)\times \mathrm{rand}+LB\right)\times \delta ,$$
where both α and δ are development adjustment parameters and take values in the interval (0, 1). UB and LB denote the lower bound and upper bound of a given problem, respectively.

#### 2.3.4. Walk and Grab Prey (Narrowed Exploitation)

When Aquila is close to the prey, it lands and attacks the prey according to the movement of the prey. This is Aquila’s final attack on the prey’s final location, and the method can be expressed as
(23)$$\begin{array}{c} x_{ij}^{t+1}=QF\times {gbest}_{j}-\left(G_{1}\times x_{ij}^{t}\times \mathrm{rand}\right)-G_{2}\times Levy\left(D\right)+rand\times G_{1} \end{array},$$
where QF represents the quality function used to balance the search strategy, and its calculation formula is shown in Equation (24). $G_{1}$ is used to represent the various actions that Aquila makes when tracking the escaping prey, which can be calculated by Equation (25). The decreasing value of $G_{2}$ from 2 to 0 during the iterative process represents the flight slope of Aquila tracking the prey when the prey is escaping, and this parameter is expressed by Equation (26).
(24)$$QF\left(t\right)=t^{\frac{2\times rand-1}{(1-T_{max}{)}^{2}}},$$
(25)$$G_{1}=2\times rand-1$$
(26)$$G_{2}=2\times \left(1-\frac{t}{T_{max}}\right),$$

### 2.4. Opposition-Based Learning (OBL)

The opposition-based learning strategy mainly compares the fitness of the current solution and its opposite solution and selects the better of the two to enter the next stage. As OBL is widely applied to the optimization of metaheuristic algorithms, researchers successively proposed variants of OBL, such as quasi OBL [42], binary student OBL [43], and specular reflection learning [44]. Basic OBL is defined as follows:

**Definition 1.** *Opposite numbers. When *x *is a real number, and* x∈[lb,ub], *the opposite number* $\tilde{x}$*is shown as*(27)$$\tilde{x}=lb+ub-x,$$

**Definition 2.** *Opposite points. When point* $P(x_{1},x_{2},\cdot \cdot \cdot ,x_{n})$ *is a point in* n*-dimensional coordinates,* $x_{1},x_{2},\cdot \cdot \cdot ,x_{n}$ *are real numbers, and* $x_{1},x_{2},\cdot \cdot \cdot ,x_{n}\in [{lb}_{i},{ub}_{i}]$; *the coordinates in the opposite point* $\tilde{P}$ *can be shown as*(28)$${\tilde{x}}_{i}={lb}_{i}+{ub}_{i}-x_{i},$$

## 3. Proposed Hybrid Algorithm (AOBLMOA)

The proposed hybrid algorithm (AOBLMOA) is based on the mayfly algorithm, which preserves the position movement method when the fitness of the male mayfly is better than the global optimal fitness method, the position movement method when the fitness of the female mayfly is better than the fitness of the corresponding male mayfly, and the mating method of the male and female mayflies to produce offspring. Then, the mating dance of the male mayflies and the random walk of the female mayflies were replaced with the method adopted in the AO, and the genetic mutation behavior of the offspring mayflies was replaced with reverse learning behavior.

### 3.1. Movement of Male Mayflies

Since female mayflies move toward male mayflies, and the birth of their offspring is also directly influenced by male mayflies, it is crucial to increase the search efficiency of male mayflies. In the process of moving the position of the male mayfly, if the fitness of the male mayfly is equal to or worse than the global optimal fitness, it means that its optimization effect in the last iteration process is not good, and it has not reached a better position than the global optimal individual mayfly. In this situation, the algorithm should make the male mayfly approach the global optimal mayfly and look for a better position in the process. The basic MOA uses the mating dance behavior to move the male mayfly at this stage. However, in real experiments, the optimization of mating dance behavior is not particularly effective.

In the AO, when $t\leq \frac{2}{3}T_{max}$, the algorithm lies in the exploration phase, and, when $t>\frac{2}{3}T_{max}$, the algorithm is in the exploitation phase. The contour flight with the short glide attack and walk and grab prey methods in the AO are the narrow search method in the algorithm exploration process and the narrow search method in the exploitation process, respectively, both of which allow Aquila to more directly rush to the prey. This takes place under the same requirement that male mayflies are equal to or worse than the global optimal solution in the MOA. Therefore, we introduce the contour flight with short glide attack method and the walk and grab prey method in the AO to replace the mating dance method in the original algorithm, to promote the male mayfly to approach the global optimal mayfly. The improved mathematical model of the movement process of the male mayfly population is as follows:(29)$$x_{ij}^{t+1}=\left\{\begin{array}{c} x_{ij}^{t}+{g*v}_{ij}^{t}+a_{1}e^{-\beta r_{p}^{2}}\left({pbest}_{ij}-x_{ij}^{t}\right)+a_{2}e^{-\beta r_{g}^{2}}\left({gbest}_{j}-x_{ij}^{t}\right)\\ ,f\left({gbest}_{j}\right)>f\left(x_{i}\right)\\ {gbest}_{j}\times \mathrm{Levy}\left(D\right)+y_{R}^{t}+\left(f_{1}-f_{2}\right)\times rand\\ ,f\left({gbest}_{j}\right)\leq f\left(x_{i}\right)||t\leq \frac{2}{3}T_{max}\\ QF\times {gbest}_{j}-\left(G_{1}\times y_{ij}^{t}\times \mathrm{rand}\right)-G_{2}\times Levy\left(D\right)+rand\times G_{1}\\ ,f\left({gbest}_{j}\right)\leq f\left(x_{i}\right)||t>\frac{2}{3}T_{max} \end{array}\right.,$$

### 3.2. Movement of Female Mayflies

In the basic MOA, when the fitness of the female mayfly is worse than that of the corresponding male mayfly, the female mayfly is attracted by the male mayfly and moves to the position of the male mayfly. When the fitness of individual female mayflies is better than that of their male counterparts, the female randomly walks because it is not attracted. Although the random walk behavior helps the female mayfly to expand the search range to a certain extent and prevents it from falling into a local optimum, in actual experiments, this method does not have a benefit effect on the optimization of the function. We believe that female mayflies, as an attracted population, should be attracted to the globally optimal individual and move according to the globally optimal position if the individual cannot be attracted to the corresponding male.

The high soar with vertical stoop method and the low flight with slow descent attack method in the AO are an expanded method in the exploration process and an expanded method in the exploitation process, respectively. Both methods are employed in the AO to expand the search range of Aquila. This not only has a similar effect as the random walk behavior performed by female mayflies but also more closely fits with the view that female mayflies are attracted to the global optimal individual when they are not attracted by male mayflies. Therefore, the improved mathematical model of the female mayfly’s positional movement process is as follows:(30)$$y_{ij}^{t+1}=\left\{\begin{array}{c} y_{ij}^{t}+{g*v}_{ij}^{t}+a_{3}e^{-\beta r_{mf}^{2}}\left(x_{ij}^{t}-y_{ij}^{t}\right)\\ ,f\left(y_{i}\right)>f\left(x_{i}\right)\\ {gbest}_{j}\times \left(1-\frac{t}{T_{max}}\right)+\left(x_{M}^{t}-{gbest}_{j}\times rand\right)\\ ,f\left(y_{i}\right)\leq f\left(x_{i}\right)||t\leq \frac{2}{3}T_{max}\\ \left({gbest}_{j}-x_{M}^{t}\right)\times \alpha -rand+\left(\left(UB-LB\right)\times \mathrm{rand}+LB\right)\times \delta \\ ,f\left(y_{i}\right)\leq f\left(x_{i}\right)||t>\frac{2}{3}T_{max} \end{array}\right.,$$

### 3.3. Stochastic OBL of Offspring Mayflies

Although OBL can perform well for most functions, for symmetric functions, the feasible solution without OBL has the same fitness as the opposite solution using OBL, which leads to the poor optimization effect of OBL in this kind of function and problem. To solve this problem, we introduce stochastic OBL, which applies a random perturbation on the basis of the OBL strategy to increase its randomness. The stochastic OBL strategy is defined as follows:

**Definition 3.** *Stochastic opposite point. On the basis of the opposite point, take a random perturbation to* $x_{i}$:
(31)$${\tilde{x}}_{i}={lb}_{i}+{ub}_{i}-x_{i}\times r,$$*where* r *represents a random value in the (0,1) interval that conforms to a Gaussian distribution.*

Introducing the stochastic OBL into the process of optimizing the offspring mayfly certainly helps to improve the performance of the algorithm by diversifying the offspring population. However, if this method is directly used on the offspring mayfly on the basis of the original algorithm, the overall efficiency of the algorithm decreases due to the increase in the time complexity. Therefore, we replace the original genetic mutation behavior of offspring mayflies with the stochastic OBL, which enables the embedding of the strategy in the algorithm without affecting the time complexity. The process of optimizing offspring mayflies using stochastic optimization is as follows:(32)$${offspring}_{ij}^{t}=\left\{\begin{array}{c} {offspring}_{ij}^{t},f\left({offspring}_{ij}^{t}\right)\leq f\left({\tilde{offspring}}_{ij}^{t}\right)\\ {\tilde{offspring}}_{ij}^{t},f\left({offspring}_{ij}^{t}\right)>f\left({\tilde{offspring}}_{ij}^{t}\right) \end{array}\right.,$$
where ${\tilde{offspring}}_{ij}^{t}$ represents the individual optimized using the stochastic OBL strategy, and ${offspring}_{ij}^{t}$ represents the individual not optimized using the stochastic OBL strategy.

### 3.4. Sensitivity Analysis

The parametric sensitivity analysis method of Xu et al. [45] is applied to AOBLMOA. Based on this, the weight minimization of a speed reducer problem in CEC2020RW problems is selected in this section for the sensitivity analysis of five core parameters in AOBLMOA: $a_{1}$, $a_{2}$, $a_{3}$, α, and δ. The details of the selected problem are shown in Section 4.3.1. And the ranges of parameters are $a_{1}\in [0.8,1.0]$, $a_{2}\in [1.3,1.5]$, $a_{3}\in [1.3,1.5]$, α∈[0.08,0.1], and δ∈[0.08,0.1]. The problem size is set to 7, and the number of iterations is set to 10,000. The average value obtained after 25 independent runs of the algorithm with different parameter combinations is shown in Table 2.

From Table 2, we can see that AOBLMOA obtains the best function value in Scenario 32, that is, $a_{1}=1.0$, $a_{2}=1.5$, $a_{3}=1.5$, α=0.1, and δ=0.1. Therefore, in the subsequent experiments, we set the parameters to be the same as those of Scenario 32.

### 3.5. Pseudocode of AOBLMOA

The pseudo-code for AOBLMOA is shown in Algorithm 1.
**Algorithm 1:** Pseudo-code of AOBLMOAInput the Initialization parameters of AOBLMOAObjective function f(x),x =${\left(x_{1},\ldots ,x_{d}\right)}^{T}$ Initialize the male mayfly population $x_{i}$(i = 1,2,…,N) Initialize the male mayfly velocities $v_{mi}$ Initialize the female mayfly population $y_{i}$(i = 1,2,…,N) Initialize the female mayfly velocities $v_{fi}$ Evaluate solutions Find global best gbest**while **t≤maxIter** do**    **for** each female $y_{i}$ **do**        **if** $f\left(y_{i}\right)\leq f\left(x_{i}\right)$ **then**           Update the $v_{fi}$ using Equation (8)           Update the location vector $y_{i}$ using Equation (7)        **else if** $f\left(y_{i}\right)>f\left(x_{i}\right)\left|\right|t\leq \frac{2}{3}T_{max}$ **then**           Update the location vector $y_{i}$ using Equation (14)        **else if** $f\left(y_{i}\right)>f\left(x_{i}\right)\left|\right|t>\frac{2}{3}T_{max}$ **then**           Update the location vector $y_{i}$ using Equation (23)        **end if**    **end for**    **for** each male $x_{i}$ **do**        **if** $f\left(x_{i}\right)\leq f\left(gbest\right)$ **then**           Update the $v_{mi}$ using Equation (2)           Update the location vector $x_{i}$ using Equation (1)        **else if** $f\left(x_{i}\right)>f\left(gbest\right)\left|\right|t\leq \frac{2}{3}T_{max}$ **then**           Update the location vector $x_{i}$ using Equation (12)        **else if** $f\left(x_{i}\right)>f\left(gbest\right)\left|\right|t>\frac{2}{3}T_{max}$ **then**           Update the location vector $x_{i}$ using Equation (22)        **end if**    **end for**    Generate the offspring mayfly population $z_{i}$ using Equations (9) and (10)    **for** each offspring $z_{i}$ **do**        Calculate the opposite solution using Equation (31)        Update the location vector $z_{i}$ using Equation (32)    **end for**    **return gbest****end while**Output the Global optimal solution **gbest**

## 4. Experimental Results and Analysis

To demonstrate AOBLMOA’s effectiveness, stability, and excellence, we first test it using 19 benchmark functions. The basic MOA and basic AO are compared with PSO, GA, GWO, the SSA, SCA, and other traditional or state-of-art metaheuristic algorithms in the classical benchmark function, and the comparison results show that the two algorithms are superior to these traditional algorithms. Based on this, we compare AOBLMOA with the basic MOA, the basic AO, AMOA combining the MOA and the AO, OBLMOA combining the MOA and the OBL strategy, and OBLAO combining the AO and the OBL strategy to prove that AOBLMOA not only outperforms classical metaheuristic algorithms but also is the optimal algorithm among the multiple algorithm combinations used in this paper. Not only that, we use CEC2017 bound constrained numerical optimization problems to compare AOBLMOA with the state-of-the-art algorithm to prove that AOBLMOA can not only be applied in numerical optimization problems but also has advantages compared with other state-of-the-art algorithms. Finally, we apply AOBLMOA to CEC2020 real-world constraint optimization problems and compare it with the three top-performing algorithms in this competition to prove that AOBLMOA is equally applicable in real-world engineering problems.

In the CEC2017BC test suite, the state-of-the-art algorithms compared with AOBLMOA are the Reptile Search Algorithm (RSA) [46], the Annealed Locust Algorithm (SGOA) [47], the Multi-Strategy Enhanced Salmon Optimization Algorithm (ESSA) [48], the improved Moth-to-Flame Algorithm (LGCMFO) [49] and the Chaos-Based Mayfly Algorithm (COLMA) [4]. The data used in the comparison are all taken from the data disclosed in the papers of each algorithm. In the CEC2020RW test suite, the introduced comparison algorithms include Self-Adaptive Spherical Search (SASS) [50], the Modified Matrix Adaptation Evolution Strategy (sCMAgES) [51], and COLSHADE [52].

The experiments are conducted using a computer Core i7-1165G7 with 16 GB RAM and 64-bit for Microsoft Windows 11. The source code is implemented using MATLAB (R2021b).

### 4.1. Benchmark Function

We use 19 benchmark functions to test the abilities of AOBLMOA to search for the global optimal solution in the problem space and to jump out of local optimal solutions and to prove the superiority of AOBLMOA compared to the original MOA, the AO, and variant algorithms. The 19 benchmark functions include the unimodal function ($f_{1}$–$f_{4}$) used to test the global search ability of the algorithm, the multimodal function ($f_{5}$–$f_{10}$) used to test the local search ability of the algorithm in more complex cases and the ability to escape the local optimum, and the fixed-dimension function ($f_{11}$–$f_{19}$) used to test the exploration ability of the algorithm in low-dimensional space. To evaluate the development ability of AOBLMOA in different problem dimensions, we set the problem dimensions of the unimodal and multimodal functions to 10, 30, 50, and 100 dimensions and analyzed the results in different dimensions. As we found in the literature [22,23,24,25,26,27,28,29,30,31,32,33,34,35,36,37,38,39,40,41], the number of iterations of each algorithm for each function is 200–100,000, and the population is 20–50. Therefore, to ensure the stability of the algorithm, each algorithm independently runs 30 times for each function, and the maximum number of iterations each time is 1000. The mathematical expression of each benchmark function is shown in Table 3.

#### 4.1.1. Convergence Analysis

Convergence analysis is the most basic step in the process of analyzing an algorithm. In this section, to verify the performance of AOBLMOA, we analyze its convergence behavior in the iterative process through a visualization method. Figure 1 lists the 2D image of the benchmark function (column 1), the search history of the algorithm in the problem space (column 2), the convergence trajectory of the algorithm (column 3), the change in the average fitness value (column 4), and the convergence curve (column 5).

The search history in the second column of Figure 1 shows the position distribution of the search factor in AOBLMOA for each iteration of the different functions. For the unimodal function, most of the search factors converge so close to the optimal point that the points in the scatter plot look sparse. For the multimodal function and the fixed-dimension function, the search factors of AOBLMOA in $f_{5}$, $f_{6}$, $f_{7}$, $f_{8}$, $f_{9}$, $f_{10}$, $f_{14}$, and $f_{16}$ can be searched around the optimal point, while, in $f_{11}$, $f_{12}$, $f_{13}$, $f_{15}$, $f_{17}$, $f_{18}$, and $f_{19}$, we can see that although the search factor falls into the local optimum in the search process once, it is not affected by the local optimum and can jump out of the local optimum and develop around the optimum. The third column shows the trajectory of the male mayfly population in the first dimension of each problem. From the trajectory, it can be seen that AOBLMOA has a higher oscillation frequency in the early iteration process, indicating its strong exploration ability in the early iteration, while a lower oscillation range in the late iteration indicates that the algorithm has a strong development ability in the late iteration.

Not only that, but, from the changes in the average fitness value of the solutions in the fourth column, we can find that most of the solutions have relatively high fitness values at the beginning of the iteration. However, within 20 iterations, the average fitness value of the solution can reach a very low interval, which shows that AOBLMOA can converge to the region of the optimal solution in fewer iterations. Similarly, we can see from the convergence curve of the algorithm in the fifth column that, since the unimodal function has only one optimal point, and the convergence process is relatively simple, the convergence curve of the algorithm for the unimodal function is relatively smooth; since the multimodal function may have multiple optimal points, and its convergence process is complicated, it may be necessary to find the global optimal point by jumping out of the local optimum. Therefore, the convergence curve of AOBLMOA for the multimodal function is similar to the stepwise convergence curve; even so, AOBLMOA finds the global optimal point in both $f_{6}$ and $f_{8}$ within single-digit iterations. For fixed-dimension functions, AOBLMOA can also find the global optimum within very few iterations.

To further analyze the convergence of AOBLMOA, we compare the convergence curves of the MOA, the AO, AMOA, OBLMOA, OBLAO, and AOBLMOA in the same graph. The parameter settings of each algorithm are shown in Table 4. Figure 2 is a comparison of the convergence curves of multiple algorithms for different functions, covering the optimal solution of each algorithm at each iteration number. We compare the MOA, the AO, and AMOA without the OBL strategy with the OBLMOA, OBLAO, and AOBLMOA with the OBL strategy and find that the algorithms that adopted the OBL strategy show a more obvious decay rate for the three functions, especially for $f_{11}$–$f_{17}$. We find that the AO easily falls into a local optimum for these functions, and OBLAO with the OBL strategy has a faster convergence speed than the basic AO, which shows that the OBL strategy helps the algorithm with iterative convergence in time. At the same time, to confirm that the fusion of the AO and the MOA is helpful to the convergence of the algorithm, we compare the AO, the MOA, and AMOA as well as OBLAO, OBLMOA, and AOBLMOA. From the figure, we find that AMOA adopting the fusion strategy has a faster convergence speed than the AO and the MOA for most functions. Similarly, AOBLMOA has better convergence than OBLAO and OBLMOA. This shows that the fusion of the AO and the MOA also helps to improve the convergence of the algorithm. Finally, comparing AOBLMOA with other algorithms, it is found that the proposed AOBLMOA converges faster than the MOA and the AO and does not fall into a local optimum. Compared with other hybrid algorithms, AOBLMOA is also the one with the fastest convergence speed and can find the global optimum in only a small number of iterations.

#### 4.1.2. Search Capability Analysis

The search capability analysis of the metaheuristic algorithm can be carried out by analyzing the development ability and the exploration ability of the algorithm. Table 5 and Table 6 show the calculation results of different algorithms for each benchmark function, including the best solution, median solution, worst solution, mean solution, and standard deviation after 30 independent runs, and the best results are shown in bold.

Since the unimodal function has only one global optimal solution, it is often used to test the exploitability of the algorithm. The optimization results of the unimodal function ($f_{1}$–$f_{4}$) in Table 5 confirm the superiority of the proposed AOBLMOA exploitation performance because its best value, median value, worst value, and mean value for the unimodal function are superior to those of other algorithms, and all reach the theoretical optimal value of the function. Moreover, the minimum standard deviation obtained by AOBLMOA also proves its superior exploitation performance with high reliability.

The multimodal function for the benchmark function is often used to evaluate the exploration ability of the algorithm. We applied the high-dimensional multimodal function ($f_{5}$–$f_{10}$) and the fixed-dimension multimodal function ($f_{11}$–$f_{19}$) to test the ability of the algorithm. Table 5 and Table 6 list the best value, median value, worst value, mean value, and standard deviation of each algorithm for high-dimensional multimodal functions and fixed-dimension multimodal functions, respectively. The results show that AOBLMOA provides the minimum metric values fir all 15 multimodal functions and reaches the theoretical optimal values for all the other functions, except $f_{5}$, $f_{7}$, $f_{9}$, and $f_{10}$. This shows that AOBLMOA also has excellent exploration capabilities.

#### 4.1.3. Stability Analysis

To evaluate the ability and stability of AOBLMOA when dealing with problems of different dimensions, we set the problem dimensions of the 10 functions, $f_{1}–f_{10}$, in Table 3 to 30, 50, and 100 dimensions. AOBLMOA and other comparison algorithms are independently run 30 times for each function, and the number of iterations is set to 1000. As shown in Table 7, Table 8 and Table 9, the excellent performance of each algorithm for each function is reflected in its best value, median value, worst value, average value, and standard deviation. Obviously, except that the best value and median of AOBLMOA in the 30-dimensional $f_{5}$ are slightly lower than those of the AO and OBLAO, it achieves the best results compared with the other algorithms for all the indicators of each function in each dimension. At the same time, the proposed algorithm can jump out of multiple local optimal solutions for functions of different dimensions and accurately reach the global optimum after finding the effective global optimal search domain, which is enough to show that its search ability does not decline with the increase in the problem dimension.

#### 4.1.4. Time Complexity Analysis

The time complexity can describe the efficiency of the algorithm operation. Zhou et al. [5] analyzed the time complexity of the variant mayfly algorithm that they proposed, and we adopt the same idea to analyze and compare the time complexity of the MOA and AOBLMOA.

The time complexity of the basic MOA and AOBLMOA is mainly related to three parameters: the problem dimension (d), population size (N), and algorithm iteration number ($T_{max}$). In a single iteration, the time complexity of each algorithm optimization process can be summarized as O(MOA)=O (population initialization)+O (position updating)+O (mayflies mating)+O (offspring mutation), O (AOBLMOA)=O (population initialization)+O (position updating)+O (mayflies mating)+O (offspring opposition−based learning). The detailed analysis of the time complexity of the algorithm is as follows:

In the basic MOA, the time complexity of the initial population is O(d×N), the calculation cost of the process of mayfly position change is $O(2\times T_{max}\times d\times N)$, the calculation amount of the mating of the mayflies is $O(T_{max}\times d\times N)$, and the calculation amount of the mutation of the offspring is $O(T_{max}\times d\times N)$. Therefore, the time complexity of the basic MOA is $O((d\times N)\times (1+4\times T_{max}))$.

In AOBLMOA, the time complexity of the initial population is O(d×N), the time complexity of the process of the mayfly position change adopted by combining AO is $O(2\times T_{max}\times d\times N)$, the time complexity of the mating process of mayflies is $O(T_{max}\times d\times N)$, and the time consumption of the offspring mayflies in stochastic OBL processes is $O(T_{max}\times d\times N)$. Thus, the time complexity of AOBLMOA is $O((d\times N)\times (1+4\times T_{max}))$.

To sum up, the time complexity of the MOA and AOBLMOA can be expressed as O(N), which shows that AOBLMOA has no significant improvement in time complexity compared with the basic MOA.

At the same time, in order to confirm the accuracy of the time complexity analyses of the two algorithms, we list the calculation time of several algorithms fors $f_{1}$–$f_{9}$ in Table 5 and Table 6. From these data, we can see that the calculation time of the MOA and AOBLMOA is similar, which proves that the time complexity of the two algorithms is basically the same.

#### 4.1.5. Statistical Analysis

In the above analysis process, after the comprehensive consideration of the experimental results obtained after each algorithm is independently run 30 times for different functions, we conclude that the proposed AOBLMOA has superior convergence, search ability, and stability. To confirm the statistical validity of this conclusion, we use statistical methods to analyze the optimization results of each algorithm for each function. The adopted statistical methods include the Wilcoxon rank sum nonparametric statistical test [53] and the Friedman statistics test.

The Wilcoxon rank sum nonparametric statistical test mainly judges whether there is a significant difference between two groups of data by comparing the p value. If p<0.05, it means that the two groups of data have a significant difference; otherwise, there is no significant difference. Based on this, we compare the optimization results of AOBLMOA for different functions with other algorithms. If p<0.05, there is a significant difference between AOBLMOA and the specific algorithm for the corresponding function; otherwise, there is no significant difference. The calculation results are shown in Table 10, where “+/−/=“ indicates the number of results of “significant advantage/significant disadvantage/no significant difference” between the corresponding algorithm and AOBLMOA. In Table 10, compared with other algorithms, AOBLMOA not only has no significant disadvantage but also has significant advantages for most functions.

The Friedman statistical test mainly judges the superiority of the algorithm by ranking the mean value in the data, and the lower the ranking value is, the more superior the algorithm is. The ranking formula is
(33)$$R_{j}=\frac{1}{N}\sum_{i=1}^{N}r_{i}^{j},$$
where i represents the i-th function, j represents the j-th algorithm, N represents the number of test functions, and $r_{i}^{j}$ represents the ranking of the j-th algorithm in the i-th function. The smaller the value of $R_{j}$ is, the higher the ranking of the algorithm is, and the better the performance is.

From the Friedman ranking in Table 11, we find that both the basic AO and the basic MOA are ranked in the last two positions for all algorithms, which shows that other hybrid algorithms are indeed optimized on the basis of the original algorithm. Not only that, the proposed AOBLMOA ranks first in all 49 functions, and its Friedman ranking and the final ranking are both first, which is enough to illustrate the strong performance of AOBLMOA compared with other algorithms.

Combining the two statistical analysis results, it can be seen that the conclusion that “AOBLMOA has superior convergence, search ability and stability compared with AO, MOA and other hybrid algorithms” is statistically valid.

### 4.2. CEC2017 Bound Constrained Numerical Optimization Problems

To demonstrate that the proposed AOBLMOA not only outperforms the original algorithms, the recently popular improved MOA, and the state-of-the-art algorithm but also can be applied to numerical optimization problems, we use a very challenging CEC2017BC function for testing. The CEC2017BC test functions include unimodal functions ($f_{1}$–$f_{3}$), multimodal functions ($f_{4}$–$f_{11}$), hybrid functions ($f_{12}$–$f_{20}$), and composite functions ($f_{21}$–$f_{30}$). The specific information is shown in Table 12.

In this experiment, we compare AOBLMOA with several state-of-the-art algorithms, such as the MOA, the AO, LGCMFO, RSA, the ESSA, the SGOA, and COLMA. All algorithms are independently run 30 times, the maximum number of iterations is 1000, and the population size is set to 30. Table 13 lists the average, standard deviation, Friedman ranking, and final ranking of each algorithm for each function. From the final ranking results in Table 13, we can see that the performance of AOBLMOA for the CEC2017 function set is not only better than the MOA and the AO but also than the other state-of-the-art algorithms, and the final ranking is first. This shows that AOBLMOA is feasible and superior when applied to numerical optimization problems.

### 4.3. CEC2020 Real-World Constrained Optimization Problems

In order to verify the feasibility of the proposed AOBLMOA in engineering optimization problems, we use the CEC2020 real-world constrained optimization problems to test it and compare it with the three best-performing algorithms in the CEC2020 RW competition. The three algorithms are SASS, sCMAgES, and COLSHADE, and the running results of these three algorithms can be found in [54]. We use a static penalty function approach to address the constraints of each problem. The standard deviation, mean value, median value, minimum value, and maximum value of each algorithm are compared after 25 independent runs for each problem. It is not difficult to see from the results in Table 14, Table 15, Table 16, Table 17, Table 18, Table 19, Table 20, Table 21, Table 22 and Table 23 that the proposed AOBLMOA is not inferior to the other three algorithms, which just shows that AOBLMOA is also feasible in real-world engineering problems.

#### 4.3.1. Weight Minimization of a Speed Reducer (WMSR)

The WMSR problem mainly describes the design of a small aircraft engine reducer. The problem contains 11 constraints and seven design variables, and the mathematical model is as follows:

minimize
$$f\left(\bar{x}\right)=0.7854x_{2}^{2}x_{1}\left(14.9334x_{3}-43.0934+3.3333x_{3}^{2}\right)+0.7854\left(x_{5}x_{7}^{2}+x_{4}x_{6}^{2}\right)-1.508x_{1}\left(x_{7}^{2}+x_{6}^{2}\right)+7.477\left(x_{7}^{3}+x_{6}^{3}\right),$$
subject to
$$g_{1}(\bar{x})=-x_{1}x_{2}^{2}x_{3}+27\leq 0,\;g_{2}(\bar{x})=-x_{1}x_{2}^{2}x_{3}^{2}+397.5\leq 0,\;g_{3}(\bar{x})=-x_{2}x_{6}^{4}x_{3}x_{4}^{-3}+1.93\leq 0,\;g_{4}(\bar{x})=-x_{2}x_{7}^{4}x_{3}x_{5}^{-3}+1.93\leq 0,\;g_{5}(\bar{x})=10x_{6}^{-3}\sqrt{16.91\times 10^{6}+{\left(745x_{4}x_{2}^{-1}x_{3}^{-1}\right)}^{2}}-1100\leq 0,\;g_{6}(\bar{x})=10x_{7}^{-3}\sqrt{157.5\times 10^{6}+{\left(745x_{5}x_{2}^{-1}x_{3}^{-1}\right)}^{2}}-850\leq 0,\;g_{7}\left(\bar{x}\right)=x_{2}x_{3}-40\leq 0,\;g_{8}(\bar{x})=-x_{1}x_{2}^{-1}+5\leq 0,\;g_{9}(\bar{x})=x_{1}x_{2}^{-1}-12\leq 0,\;g_{10}(\bar{x})=1.5x_{6}-x_{4}+1.9\leq 0,\;g_{11}(\bar{x})=1.1x_{7}-x_{5}+1.9\leq 0,\;$$
with bounds
$$\begin{array}{c} 0.7\leq x_{2}\leq 0.8,17\leq x_{3}\leq 28,2.6\leq x_{1}\leq 3.6,\\ 5\leq x_{7}\leq 5.5,7.3\leq x_{5},x_{4}\leq 8.3,2.9\leq x_{6}\leq 3.9, \end{array}$$

#### 4.3.2. Optimal Design of Industrial Refrigeration System (ODIRS)

The ODRIS problem is an optimization problem for an industrial refrigeration system. It contains 15 constraints and 14 design variables. Its specific mathematical model is as follows:

minimize
$$\begin{array}{c} f(\bar{x})=63098.88x_{2}x_{4}x_{12}+5441.5x_{2}^{2}x_{12}+115055.5x_{2}^{1.664}x_{6}+6172.27x_{2}^{2}x_{6}\\ +63098.88x_{1}x_{3}x_{11}+5441.5x_{1}^{2}x_{11}+115055.5x_{1}^{1.664}x_{5}+6172.27x_{1}^{2}x_{5}\\ +140.53x_{1}x_{11}+281.29x_{3}x_{111}+70.26x_{1}^{2}+281.29x_{1}x_{3}+281.29x_{3}^{2}\\ +14437x_{8}^{1.8812}x_{12}^{0.3424}x_{10}x_{14}^{-1}x_{1}^{2}x_{7}x_{9}^{-1}+20470.2x_{7}^{2.893}x_{11}^{0.316}x_{1}^{2}, \end{array}$$
subject to
$$g_{1}(\bar{x})=1.524x_{7}^{-1}\leq 1,\;g_{2}(\bar{x})=1.524x_{8}^{-1}\leq 1,\;g_{3}(\bar{x})=0.07789x_{1}-2x_{7}^{-1}x_{9}-1\leq 0,\;g_{4}(\bar{x})=7.05305x_{9}^{-1}x_{1}^{2}x_{10}x_{8}^{-1}x_{2}^{-1}x_{14}^{-1}-1\leq 0,\;g_{5}(\bar{x})=0.0833x_{13}^{-1}x_{14}-1\leq 0,\;g_{6}(\bar{x})=47.136x_{2}^{0.333}x_{10}^{-1}x_{12}-1.333x_{8}x_{13}^{2.1195}+62.08x_{13}^{2.1195}x_{12}^{-1}x_{8}^{0.2}x_{10}^{-1}-1\leq 0,\;g_{7}(\bar{x})=0.04771x_{10}x_{8}^{1.8812}x_{12}^{0.3424}-1\leq 0,\;g_{8}(\bar{x})=0.0488x_{9}x_{7}^{1.893}x_{11}^{0.316}-1\leq 0,\;g_{9}(\bar{x})=0.0099x_{1}x_{3}^{-1}-1\leq 0,\;g_{10}(\bar{x})=0.0193x_{2}x_{4}^{-1}-1\leq 0,\;g_{11}(\bar{x})=0.0298x_{1}x_{5}^{-1}-1\leq 0,\;g_{12}(\bar{x})=0.056x_{2}x_{6}^{-1}-1\leq 0,\;g_{13}(\bar{x})=2x_{9}^{-1}-1\leq 0,\;g_{14}(\bar{x})=2x_{10}^{-1}-1\leq 0,\;g_{15}(\bar{x})=x_{12}x_{11}^{-1}-1\leq 0,$$
with bounds
$$0.001\leq x_{i}\leq 5,i=1,\ldots ,14,$$

#### 4.3.3. Tension/Compression Spring Design (TCSD Case 1)

The TCSD1 problem is a relatively classic engineering optimization problem, and many metaheuristic algorithms use this problem to prove its feasibility in engineering optimization problems. The main objective of this problem is to optimize the weight of a tension or compression spring. It consists of 4 constraints and 3 design variables: wire diameter ($x_{1}$), mean coil diameter ($x_{2}$), and number of coils ($x_{3}$). The mathematical model of the problem appears as follows:

minimize
$$f\left(\bar{x}\right)=\left(x_{3}+2\right)x_{2}{x_{1}}^{2},$$
subject to
$$g_{1}\left(\bar{x}\right)=1-\frac{{x_{2}}^{3}x_{3}}{71785{x_{1}}^{4}}\leq 0,\;g_{2}\left(\bar{x}\right)=\frac{4{x_{2}}^{2}-x_{1}x_{2}}{12566(x_{2}{x_{1}}^{3}-{x_{1}}^{4})}+\frac{1}{5108{x_{1}}^{2}}\leq 0,\;g_{3}\left(\bar{x}\right)=1-\frac{140.45x_{1}}{{x_{2}}^{2}x_{3}}\leq 0,\;g_{4}\left(\bar{x}\right)=\frac{x_{1}+x_{2}}{1.5}-1\leq 0,$$
with bounds
$$0.05{\leq x}_{1}\leq 2,\;0.25{\leq x}_{2}\leq 1.3,2{\leq x}_{3}\leq 15,$$

#### 4.3.4. Multiple Disk Clutch Brake Design Problem (MDCBDP)

The MDCBDP problem is described by nine constraints and five integer decision variables, and its main purpose is to minimize the mass of the multiplate clutch brake. The decision variables for this problem include the inner radius ($x_{1}$), outer radius ($x_{2}$), disc thickness ($x_{3}$), actuator force ($x_{4}$), and number of friction surfaces ($x_{5}$). Its mathematical model is as follows:

minimize
$$f\left(\bar{x}\right)=\pi \left(x_{2}^{2}-x_{1}^{2}\right)x_{3}\left(x_{5}+1\right)\rho ,$$
subject to
$$g_{1}(\bar{x})=-p_{max}+p_{rz}\leq 0,\;g_{2}(\bar{x})=p_{rz}V_{sr}-V_{sr,max}p_{max}\leq 0,\;g_{3}(\bar{x})=\Delta R+x_{1}-x_{2}\leq 0,\;g_{4}(\bar{x})=-L_{max}+\left(x_{5}+1\right)\left(x_{3}+\delta \right)\leq 0,\;g_{5}(\bar{x})=sM_{s}-M_{h}\leq 0,\;g_{6}(\bar{x})=T\geq 0,\;g_{7}(\bar{x})=-V_{sr,\;\max\;}+V_{sr}\leq 0,\;g_{8}\left(\bar{x}\right)=T-T_{max}\leq 0,$$
where
$$\begin{array}{c} M_{h}=\frac{2}{3}\mu x_{4}x_{5}\frac{x_{2}^{3}-x_{1}^{3}}{x_{2}^{2}-x_{1}^{2}}N\cdot \mathrm{mm}\\ \omega =\frac{\pi n}{30}\mathrm{rad}/s\\ A=\pi \left(x_{2}^{2}-x_{1}^{2}\right){\mathrm{mm}}^{2}\\ p_{rz}=\frac{x_{4}}{A}N/{\mathrm{mm}}^{2}\\ V_{sr}=\frac{\pi R_{sr}n}{30}\mathrm{mm}/s,\\ R_{sr}=\frac{2}{3}\frac{x_{2}^{3}-x_{1}^{3}}{x_{2}^{2}x_{1}^{2}}\mathrm{mm},\\ T=\frac{I_{z}\omega }{M_{h}+M_{f}},\\ \Delta R=20\;\mathrm{mm},L_{max}=30\;\mathrm{mm},\mu =0.6\\ V_{sr,max}=10\;m/s,\delta =0.5\;\mathrm{mm},s=1.5\\ T_{max}=15\;s,n=250\;\mathrm{rpm},I_{z}=55\;\mathrm{Kg}\cdot m^{2},\\ M_{s}=40\;\mathrm{Nm},M_{f}=3\;\mathrm{Nm},\;\mathrm{and}\;p_{max}=1. \end{array}$$
with bounds
$$60\leq x_{1}\leq 80,90\leq x_{2}\leq 110,1\leq x_{3}\leq 3,0\leq x_{4}\leq 1000,2\leq x_{5}\leq 9.$$

#### 4.3.5. Planetary Gear Train Design Optimization (PGTDO)

The main goal of PGTDO is to minimize the maximum error of the car transmission ratio by calculating the number of gear teeth in the automatic planetary transmission system. The mathematical model of the problem is shown below, with six integer variables and 11 constraints.

Minimize
$$f\left(\bar{x}\right)=max\left|i_{k}-i_{0k}\right|,k=\left\{1,2,..,R\right\},$$
where
$$\begin{array}{c} i_{1}=\frac{N_{6}}{N_{4}},i_{01}=3.11,i_{2}=\frac{N_{6}\left(N_{1}N_{3}+N_{2}N_{4}\right)}{N_{1}N_{3}\left(N_{6}-N_{4}\right)},i_{0R}=-3.11\\ I_{R}=-\frac{N_{2}N_{6}}{N_{1}N_{3}},i_{02}=1.84,\bar{x}=\left\{p,N_{6},N_{5},N_{4},N_{3},N_{2},N_{1},m_{2},m_{1}\right\}, \end{array}$$
subject to
$$g_{1}(\bar{x})=m_{3}\left(N_{6}+2.5\right)-D_{max}\leq 0,\;g_{2}(\bar{x})=m_{1}\left(N_{1}+N_{2}\right)+m_{1}\left(N_{2}+2\right)-D_{max}\leq 0,\;g_{3}(\bar{x})=m_{3}\left(N_{4}+N_{5}\right)+m_{3}\left(N_{5}+2\right)-D_{max}\leq 0,\;g_{4}(\bar{x})=\left|m_{1}\left(N_{1}+N_{2}\right)-m_{3}\left(N_{6}-N_{3}\right)\right|-m_{1}-m_{3}\leq 0,\;g_{5}(\bar{x})=-\left(N_{1}+N_{2}\right)\sin(\pi /p)+N_{2}+2+\delta_{22}\leq 0,\;g_{6}(\bar{x})=-\left(N_{6}-N_{3}\right)\sin(\pi /p)+N_{3}+2+\delta_{33}\leq 0,\;g_{7}(\bar{x})=-\left(N_{4}+N_{5}\right)\sin(\pi /p)+N_{5}+2+\delta_{55}\leq 0,\;\begin{array}{c} g_{8}(\bar{x})={\left(N_{3}+N_{5}+2+\delta_{35}\right)}^{2}-{\left(N_{6}-N_{3}\right)}^{2}-{\left(N_{4}+N_{5}\right)}^{2}\\ +2\left(N_{6}-N_{3}\right)\left(N_{4}+N_{5}\right)cos\left(\frac{2\pi }{p}-\beta \right)\leq 0, \end{array}\;g_{9}(\bar{x})=N_{4}-N_{6}+2N_{5}+2\delta_{56}+4\leq 0,\;g_{10}(\bar{x})=2N_{3}-N_{6}+N_{4}+2\delta_{34}+4\leq 0,\;h_{1}(\bar{x})=\frac{N_{6}-N_{4}}{p}=\mathrm{integer},\;\delta_{22}=\delta_{33}=\delta_{55}=\delta_{35}=\delta_{56}=0.5,\;\beta =\frac{\cos^{-1}\left({\left(N_{4}+N_{5}\right)}^{2}+{\left(N_{6}-N_{3}\right)}^{2}-{\left(N_{3}+N_{5}\right)}^{2}\right)}{2\left(N_{6}-N_{3}\right)\left(N_{4}+N_{5}\right)},D_{max}=220,$$
with bounds
$$\begin{array}{c} p=(3,4,5)\\ m_{1}=(1.75,2.0,2.25,2.5,2.75,3.0),\\ m_{3}=(1.75,2.0,2.25,2.5,2.75,3.0),\\ 17\leq N_{1}\leq 96,14\leq N_{2}\leq 54,14\leq N_{3}\leq 51\\ 17\leq N_{4}\leq 46,14\leq N_{5}\leq 51,48\leq N_{6}\leq 124,\\ \mathrm{and}\;N_{i}\;=\;\mathrm{integer}. \end{array}$$

#### 4.3.6. Hydro-Static Thrust-Bearing Design Problem (HTBDP)

The HTBDP problem is primarily about optimizing bearing power losses by using four design variables: oil viscosity, bearing radius, flow rate, and groove radius. In addition to the above four design variables, the problem also contains seven nonlinear constraints, and its mathematical model is as follows:

minimize
$$f\left(\bar{x}\right)=\frac{QP_{0}}{0.7}+E_{f},$$
subject to
$$g_{1}(\bar{x})=1000-P_{0}\leq 0,\;g_{2}(\bar{x})=W-101000\leq 0,\;g_{3}(\bar{x})=5000-\frac{W}{\pi \left(R^{2}-R_{0}^{2}\right)}\leq 0,\;g_{4}(\bar{x})=50-P_{0}\leq 0,\;g_{5}(\bar{x})=0.001-\frac{0.0307}{386.4P_{0}}\left(\frac{Q}{2\pi Rh}\right)\leq 0,\;g_{6}(\bar{x})=R-R_{0}\leq 0,\;g_{7}(\bar{x})=h-0.001\leq 0,$$
where
$$W=\frac{\pi P_{0}}{2}\frac{R^{2}-R_{0}^{2}}{\ln\left(\frac{R}{R_{0}}\right)},P_{0}=\frac{6\mu Q}{\pi h^{3}}\ln\left(\frac{R}{R_{0}}\right),\;E_{f}=9336Q\times 0.0307\times 0.5\Delta T,\Delta T=2\left(10^{P}-559.7\right),\;P=\frac{{log}_{10}{log}_{10}\left(8.122\times 10^{6}\mu +0.8\right)+3.55}{10.04},\;h={\left(\frac{2\pi \times 750}{60}\right)}^{2}\frac{2\pi \mu }{E_{f}}\left(\frac{R^{4}}{4}-\frac{R_{0}^{4}}{4}\right),$$
with bounds
$$1\leq R\leq 16,1\leq R_{0}\leq 16,1\times 10^{-6}\leq \mu \leq 16\times 10^{-6},1\leq Q\leq 16,$$

#### 4.3.7. Four-Stage Gear Box Problem (FGBP)

The FGBP aims at minimizing the weight of the gearbox and has 22 design variables, which are very discrete, including gear position, gear teeth number, blank thickness, etc. At the same time, the problem also has 86 nonlinear constraints. The comparison of the results of each algorithm in this problem is shown in Table 20.

#### 4.3.8. Gas Transmission Compressor Design (GTCD)

The GTCD problem contains four design variables and one constraint condition, which is used to optimize the design of the gas transmission compressor. Its mathematical model is as follows:

minimize
$$f\left(\bar{x}\right)=8.61\times 10^{5}x_{1}^{\frac{1}{2}}x_{2}x_{3}^{-\frac{2}{3}}x_{4}^{-\frac{1}{2}}+3.69\times 10^{4}x_{3}+7.72\times 10^{8}x_{1}^{-1}x_{2}^{0.219}-765.43\times 10^{6}x_{1}^{-1},$$
subject to
$$x_{4}x_{2}^{-2}+x_{2}^{-2}-1\leq 0,$$
with bounds
$$20\leq x_{1}\leq 50,1\leq x_{2}\leq 10,20\leq x_{3}\leq 50,0.1\leq x_{4}\leq 60,$$

#### 4.3.9. Tension/Compression Spring Design (TCSD Case 2)

The TCSD2 problem is mainly used to optimize the volume required for manufacturing helical compression spring steel wire. This problem mainly includes three design variables, the outer diameter ($x_{1}$), the number of spring coils ($x_{2}$) and the spring steel wire diameter ($x_{3}$), and eight nonlinear constraints. Its mathematical model is as follows:

minimize
$$f\left(\bar{x}\right)=\frac{\pi^{2}x_{2}x_{3}^{2}\left(x_{1}+2\right)}{4},$$
subject to
$$g_{1}\left(\bar{x}\right)=\frac{8000C_{f}x_{2}}{\pi x_{3}^{3}}-189000\leq 0,\;g_{2}(\bar{x})=l_{f}-14\leq 0,\;g_{3}(\bar{x})=0.2-x_{3}\leq 0,\;g_{4}(\bar{x})=x_{2}-3\leq 0,\;g_{5}(\bar{x})=3-\frac{x_{2}}{x_{3}}\leq 0,\;g_{6}(\bar{x})=\sigma_{p}-6\leq 0,\;g_{7}(\bar{x})=\sigma_{p}+\frac{700}{K}+1.05\left(x_{1}+2\right)x_{3}-l_{f}\leq 0,\;g_{8}(\bar{x})=1.25-\frac{700}{K}\leq 0,$$
where
$$C_{f}=\frac{4\frac{x_{2}}{x_{3}}-1}{4\frac{x_{2}}{x_{3}}-4}+\frac{0.615x_{3}}{x_{2}},K=\frac{11.5\times 10^{6}x_{3}^{4}}{8x_{1}x_{2}^{3}},\sigma_{p}=\frac{300}{K},l_{f}=\frac{1000}{K}+1.05\left(x_{1}+2\right)x_{3},$$
with bounds
$$1\leq x_{1}\;(\mathrm{integer})\;\leq 70,0.6\leq x_{2}\;(\mathrm{continuous})\;\leq 3,$$
$$\begin{array}{cc} x_{3}(\mathrm{discrete})\in & \{0.009,0.0095,0.0104,0.0118,0.0128,0.0132,0.014,0.015,0.0162,0.0173,0.018,0.020,\\ & 0.023,0.025,0.028,0.032,0.035,0.041,0.047,0.054,0.063,0.072,0.080,0.092,0.0105,\\ & 0.120,0.135,0.148,0.162,0.177,0.192,0.207,0.225,0.244,0.263,0.283,0.307,0.0331\\ & 0.362,0.394,0.4375,0.500\}, \end{array}$$

#### 4.3.10. Topology Optimization (TO)

The main purpose of this problem is to optimize the material layout for a given load set given the design search space and constraints related to system performance. The mathematical model is as follows:

minimize
$$f\left(\bar{x}\right)=U^{T}KU=\sum_{e=1}^{N}\;{\left(x_{e}\right)}^{p}u_{e}^{T}k_{0}u_{0},$$
subject to
$$h_{1}(\bar{x})=\frac{V\left(\bar{x}\right)}{V_{0}}-f=0,\;h_{2}(\bar{x})=KU-F=0,$$
with bounds
$$0<{\bar{x}}_{min}\leq x\leq 1.$$

## 5. Conclusions and Future Directions

In this paper, we propose a metaheuristic algorithm that combines the MOA, the AO, and the OBL strategies, namely, AOBLMOA. The algorithm takes the MOA as the framework, assigns the search methods of Aquila in the AO to the male and female mayfly populations in the MOA, and replaces the mutation strategy of the offspring mayfly population with the stochastic OBL strategy. To verify the effectiveness, superiority, and feasibility of the proposed algorithm for different types of problems, we successively apply AOBLMOA to 19 benchmark functions, 30 CEC2017 functions, and 10 CEC2020 real-world constrained optimization problems. From the obtained results and statistical analysis results, it can be seen that the algorithm has a great improvement compared with the original algorithm, has advantages compared with the recently proposed algorithm, and is feasible in numerical optimization and practical engineering optimization problems. However, the algorithm is designed for continuous problems, not binary or discrete problems. Therefore, AOBLMOA cannot solve discrete problems such as the TSP problem and the VRP problem.

In future work, we suggest that researchers interested in AOBLMOA can further optimize it and even design a binary, discrete, or multi-objective AOBLMOA. It is also interesting to apply AOBLMOA to large-scale applications such as neural network optimization, workshop task scheduling, robot path planning, text and data mining, image segmentation, signal denoising, oil and gas pipeline network transportation, feature selection, etc. The MATLAB codes for AOBLMOA are available at https://github.com/SheldonYnuup/AOBLMOA (accessed on 1 August 2023), to help researchers with further research.

## Figures and Tables

**Figure 1 biomimetics-08-00381-f001:**
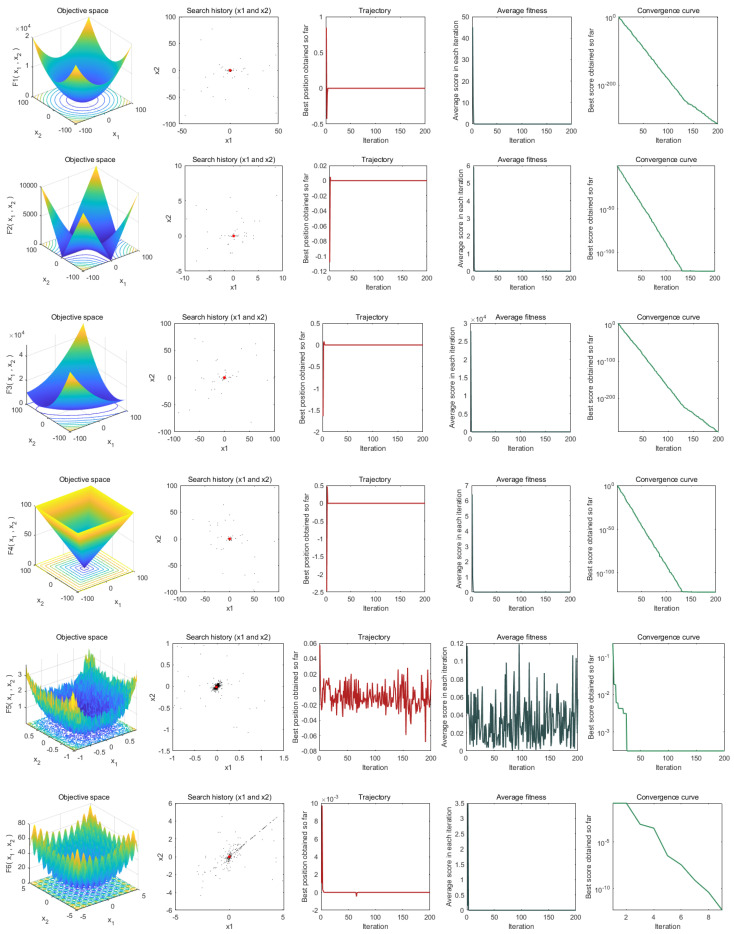
Benchmark functions.

**Figure 2 biomimetics-08-00381-f002:**
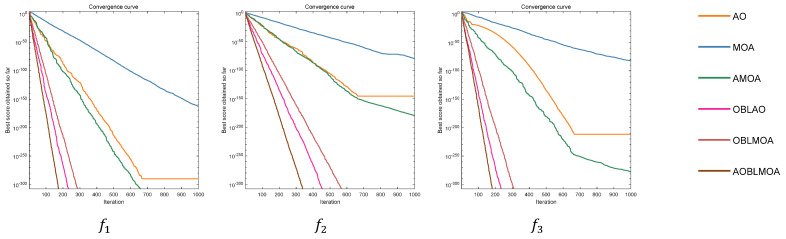
Convergence comparison among algorithms.

**Table 1 biomimetics-08-00381-t001:** The metaheuristic algorithms mentioned in the Introduction.

Type	Algorithm	Ref.	Inspiration	Characteristic
SIA	PSO	[22]	Foraging behavior of birds	The bird swarm flies to the best birds and searches in the process
	SSA	[23]	Navigation and foraging behavior of salps in the ocean	The lead salp searches first, and followers search after the leader
	GWO	[24]	The leadership hierarchy and hunting process of gray wolves	The whole pack moves toward the three best wolves
	MBO	[25]	Migration behavior of monarch butterflies	In the process of population migration, old individuals are eliminated, and new individuals are created and adapt to the environment
	BES	[26]	Bald eagles’ attack on prey	Each individual can search three times through the three methods: select, search and swoop, and stay in the optimal position
	MPA	[27]	The predation process of marine predators	MPA combines Brownian motion, Lévy flight, and other random generation strategies and uses different strategies in different stages
	WOA	[28]	The behavior of humpback whales	Whale groups search the problem space by encircling both prey and bubble nets
	MS	[29]	Navigation method of moths	Moths approach a flame by the spiral search method
EA	GA	[30]	Darwinian theory of evolution	Optimal solution candidates are filtered and continuously obtained through crossover, mutation, and selection operations
	DE	[31]	Evolutionary phenomenon	On the basis of GA, a difference operation is added to carry out the variation
	BBO	[32]	Biogeography associated with species migration	BBO incorporates the migration and mutation behavior of species
	CS	[33]	Evolution of the cuckoo	CS simulates the behavior of a cuckoo laying eggs, combining the Lévy flight and random selection methods
PhA	MVO	[34]	The concepts of black holes, white holes, and wormholes in the universe	Black holes and white holes are used for exploration and wormholes for exploitation
	EO	[35]	Simple well-mixed dynamic mass balance on a control volume	Search agents randomly update their concentration with respect to some talented particles, called equilibrium candidates, to finally reach an equilibrium state as the optimal results
	GSA	[36]	The law of gravity and mass interactions	The law of gravity and the law of motion are incorporated into the motion of particles
	LSA	[37]	The natural phenomenon of lightning	Transition projectiles, space projectiles, and lead projectiles are used to optimize problems
	AOA	[38]	The law of physics of Archimedes’ Principle	Each individual has four attributes, position, density, volume, and acceleration, and adjusts the acceleration by changing the density and volume; the acceleration and current position determine the new position
HA	TLBO	[39]	The influence of a teacher on the output of learners	TLBO is divided into two parts: the first part is the “teacher stage”, that is, learning from teachers; the second part is the “learner phase”, that is, learning through the interaction between learners
	HFA	[40]	The efforts of human society to become felicity	The population is divided into elites, disciples, and ordinary people; later, people’s minds changed in three ways: the influence of elites, personal experience, and drastic changes
	PO	[41]	The multi-phased process of politics	PO integrates party formation, constituency allocation, party elections, party switching, campaigning, and parliamentary affairs into the optimization process

**Table 2 biomimetics-08-00381-t002:** Sensitivity analysis to AOBLMOA.

Scenario	Parameter Values					Ave.
	$a_{1}$	$a_{2}$	$a_{3}$	α	δ	
1	0.8	1.3	1.3	0.08	0.08	2994.424637
2	1	1.3	1.3	0.08	0.08	2994.424557
3	0.8	1.5	1.3	0.08	0.08	2994.424646
4	1	1.5	1.3	0.08	0.08	2994.42457
5	0.8	1.3	1.5	0.08	0.08	2994.424561
6	1	1.3	1.5	0.08	0.08	2994.42453
7	0.8	1.5	1.5	0.08	0.08	2994.424518
8	1	1.5	1.5	0.08	0.08	2994.424622
9	0.8	1.3	1.3	0.1	0.08	2994.42454
10	1	1.3	1.3	0.1	0.08	2994.424568
11	0.8	1.5	1.3	0.1	0.08	2994.424538
12	1	1.5	1.3	0.1	0.08	2994.424614
13	0.8	1.3	1.5	0.1	0.08	2994.424661
14	1	1.3	1.5	0.1	0.08	2994.424638
15	0.8	1.5	1.5	0.1	0.08	2994.424518
16	1	1.5	1.5	0.1	0.08	2994.424658
17	0.8	1.3	1.3	0.08	0.1	2994.424799
18	1	1.3	1.3	0.08	0.1	2994.424532
19	0.8	1.5	1.3	0.08	0.1	2994.424554
20	1	1.5	1.3	0.08	0.1	2994.424605
21	0.8	1.3	1.5	0.08	0.1	2994.424557
22	1	1.3	1.5	0.08	0.1	2994.42454
23	0.8	1.5	1.5	0.08	0.1	2994.424548
24	1	1.5	1.5	0.08	0.1	2994.424589
25	0.8	1.3	1.3	0.1	0.1	2994.42452
26	1	1.3	1.3	0.1	0.1	2994.424547
27	0.8	1.5	1.3	0.1	0.1	2994.42457
28	1	1.5	1.3	0.1	0.1	2994.424538
29	0.8	1.3	1.5	0.1	0.1	2994.424663
30	1	1.3	1.5	0.1	0.1	2994.424617
31	0.8	1.5	1.5	0.1	0.1	2994.424755
32	1	1.5	1.5	0.1	0.1	**2994.42451**

**Table 3 biomimetics-08-00381-t003:** Benchmark functions.

Expression	Dimensions	Range	$f_{min}$
$f_{1}\left(X\right)=\sum_{i=1}^{D}x_{i}^{2}$	10, 30, 50, 100	[−100, 100]	0
$f_{2}\left(X\right)=\sum_{i=1}^{D}\left|x_{i}\right|+\prod_{i=1}^{D}\left|x_{i}\right|$	10, 30, 50, 100	[−10, 10]	0
$f_{3}\left(X\right)=\sum_{i=1}^{D}\;{\left(\sum_{j=1}^{i}\;x_{j}\right)}^{2}$	10, 30, 50, 100	[−100, 100]	0
$f_{4}\left(X\right)=\underset{i}{\max}\left\{\left|x_{i}\right|,1\leq x_{i}\leq D\right\}$	10, 30, 50, 100	[−100, 100]	0
$f_{5}\left(X\right)=\sum_{i=1}^{D}{ix}_{i}^{4}+random[0,1)$	10, 30, 50, 100	[−128, 128]	0
$f_{6}\left(X\right)=\sum_{i=1}^{D}{[x}_{i}^{2}-10\cos\left(2\pi x_{i}\right)+10]$	10, 30, 50, 100	[−5.12, 5.12]	0
$f_{7}\left(X\right)=\sum_{i=1}^{D}-20\exp\left(-0.2\sqrt{\frac{1}{D}\sum_{i=1}^{D}x_{i}^{2}}\right)-\exp\left(\frac{1}{D}\sum_{i=1}^{D}\cos\left(2\pi x_{i}\right)\right)+20+e$	10, 30, 50, 100	[−32, 32]	0
$f_{8}\left(X\right)=\frac{1}{4000}\sum_{i=1}^{D}x_{i}^{2}-\prod_{i=1}^{D}\cos\left(\frac{x_{i}}{\sqrt{i}}\right)+1$	10, 30, 50, 100	[−600, 600]	0
$f_{9}\left(X\right)=\begin{array}{c} \frac{\pi }{n}\left\{10\sin\left(\pi y_{1}\right)\right\}+\sum_{i=1}^{n-1}\;{\left(y_{i}-1\right)}^{2}\left[\begin{array}{c} 1+10\sin^{2}\left(\pi y_{i+1}\right)\\ +\sum_{i=1}^{n}\;u\left(x_{i},10,100,4\right) \end{array}\right],\\ \mathrm{where}\;y_{i}=1+\frac{x_{i}+1}{4},u\left(x_{i},a,k,m\right)\left\{\begin{array}{cc} K{\left(x_{i}-a\right)}^{m} & \;\mathrm{if}\;x_{i}>a\\ 0 & -a⩽x_{i}⩾a\\ K{\left(-x_{i}-a\right)}^{m} & -a⩽x_{i} \end{array}\right. \end{array}$	10, 30, 50, 100	[−50, 50]	0
$f_{10}\left(X\right)=0.1\left\{\sin^{2}\left(3\pi x_{i}\right)+\sum_{i=1}^{D}\begin{array}{c} {\left(x_{i}-1\right)}^{2}\left[1+\sin^{2}\left(3\pi x_{i}+1\right)\right]\\ +{\left(x_{i}-1\right)}^{2}[1+\sin^{2}\left(2\pi x_{i}\right)] \end{array}\right\}\;+\sum_{i=1}^{D}u(x_{i},5,100,4)$	10, 30, 50, 100	[−50, 50]	0
$f_{11}\left(X\right)=\sum_{i=1}^{11}{\left[a_{i}-\frac{x_{1}(b_{i}^{2}+b_{i}x_{2})}{b_{i}^{2}+b_{i}x_{3}+x_{4}}\right]}^{2}$	4	[−5, 5]	0.00030
$f_{12}\left(X\right)=4x_{1}^{2}-2.1x_{1}^{4}+\frac{1}{3}x_{1}^{6}+x_{1}x_{2}-4x_{2}^{2}+4x_{2}^{4}$	2	[−5, 5]	−1.0316
$f_{13}\left(X\right)={\left(x_{2}-\frac{5.1}{4\pi^{2}}x_{1}^{2}+\frac{5}{\pi }x_{1}-6\right)}^{2}+10\left(1-\frac{1}{8\pi }\right)cosx_{1}+10$	2	[−5, 5]	0.398
$f_{14}\left(X\right)=\left[1+{\left(x_{1}+x_{2}+1\right)}^{2}\left(19-14x_{1}+3{x_{1}}^{2}-14x_{2}+6x_{1}x_{2}+3{x_{2}}^{2}\right)\right]\times [30\;+{\left(2x_{1}-3x_{2}\right)}^{2}\times (18-32x_{1}+12{x_{1}}^{2}+48x_{2}-36x_{1}x_{2}\;+27{x_{2}}^{2})]$	2	[−2, 2]	3
$f_{15}\left(X\right)=-\sum_{i=1}^{4}c_{i}exp(-\sum_{j=1}^{3}a_{ij}{\left(x_{j}-p_{ij}\right)}^{2})$	3	[0, 1]	−3.86
$f_{16}\left(X\right)=-\sum_{i=1}^{4}c_{i}exp(-\sum_{j=1}^{6}a_{ij}{\left(x_{j}-p_{ij}\right)}^{2})$	6	[0, 1]	−3.32
$f_{17}\left(X\right)=-\sum_{i=1}^{5}\;{\left[\left(X-a_{i}\right){\left(X-a_{i}\right)}^{T}+c_{i}\right]}^{-1}$	4	[0, 1]	−10.1532
$f_{18}\left(X\right)=-\sum_{i=1}^{7}\;{\left[\left(X-a_{i}\right){\left(X-a_{i}\right)}^{T}+c_{i}\right]}^{-1}$	4	[0, 1]	−10.4028
$f_{19}\left(X\right)=-\sum_{i=1}^{10}\;{\left[\left(X-a_{i}\right){\left(X-a_{i}\right)}^{T}+c_{i}\right]}^{-1}$	4	[0, 1]	−10.5363

**Table 4 biomimetics-08-00381-t004:** Parameters of algorithms.

Parameters	MOA	AO	AMOA	OBLMOA	OBLAO	AOBLMOA
Population size	30	30	30	30	30	30
$a_{1}$, $a_{2}$$\;\mathrm{and}\;a_{3}$	1.0, 1.5, 1.5	-	1.0, 1.5, 1.5	1.0, 1.5, 1.5	-	1.0, 1.5, 1.5
g	0.9–0.4		0.9–0.4	0.9–0.4	-	0.9–0.4
α	-	0.1	0.1	-	0.1	0.1
δ	-	0.1	0.1	-	0.1	0.1

**Table 5 biomimetics-08-00381-t005:** Comparison between algorithms for $f_{1}$–$f_{10}$; dimension fixed to 10.

Function		AO	MOA	AMOA	OBLAO	OBLMOA	AOBLMOA
$f_{1}$	best	8.04 × 10^−303^	7.15 × 10^−171^	0	0	0	**0**
	median	4.74 × 10^−288^	3.48 × 10^−166^	0	0	0	**0**
	worst	4.00 × 10^−206^	2.28 × 10^−159^	4.90 × 10^−324^	0	0	**0**
	mean	1.33 × 10^−207^	8.83 × 10^−161^	0	0	0	**0**
	std	0	0	0	0	0	**0**
	time	0.025520833	0.1942708	0.2098958	0.025521	0.173438	0.146875
$f_{2}$	best	2.8314 × 10^−153^	4.334 × 10^−93^	1.92 × 10^−195^	0	0	**0**
	median	5.3136 × 10^−145^	2.295 × 10^−88^	3.42 × 10^−183^	0	0	**0**
	worst	9.4069 × 10^−102^	2.498 × 10^−79^	9.04 × 10^−158^	0	0	**0**
	mean	3.1364 × 10^−103^	1.156 × 10^−80^	3.01 × 10^−159^	0	0	**0**
	std	1.6886 × 10^−102^	4.731 × 10^−80^	0	0	0	**0**
	time	0.028125	0.2125	0.2171875	0.028646	0.186458	0.1989583
$f_{3}$	best	2.4268 × 10^−297^	7.359 × 10^−88^	4.15 × 10^−306^	0	0	**0**
	median	4.2447 × 10^−286^	3.599 × 10^−82^	2.39 × 10^−278^	0	0	**0**
	worst	1.5943 × 10^−198^	4.746 × 10^−73^	1.66 × 10^−248^	0	0	**0**
	mean	9.7774 × 10^−200^	1.582 × 10^−74^	5.53 × 10^−250^	0	0	**0**
	std	0	8.52 × 10^−74^	0	0	0	**0**
	time	0.036458333	0.2098958	0.2328125	0.054688	0.210938	0.2255208
$f_{4}$	best	2.5033 × 10^−151^	6.427 × 10^−38^	3.46 × 10^−212^	0	0	**0**
	median	1.6094 × 10^−145^	4.494 × 10^−31^	7.52 × 10^−190^	0	0	**0**
	worst	5.4551 × 10^−99^	2.419 × 10^−26^	9.09 × 10^−164^	0	0	**0**
	mean	2.2999 × 10^−100^	1.108 × 10^−27^	3.03 × 10^−165^	0	0	**0**
	std	9.9082 × 10^−100^	4.473 × 10^−27^	0	0	0	**0**
	time	0.022395833	0.2052083	0.2041667	0.025	0.180208	0.19375
$f_{5}$	best	2.02018 × 10^−6^	0.0001994	4.293 × 10^−6^	1.7 × 10^−6^	1.18 × 10^−6^	**5.63 × 10^−7^**
	median	**2.76003 × 10^−5^**	0.000643	3.137 × 10^−5^	3.49 × 10^−5^	2.99 × 10^−5^	2.931 × 10^−5^
	worst	0.000249345	0.0020499	0.0002063	0.000197	0.000234	**0.000106**
	mean	5.98305 × 10^−5^	0.0008928	4.658 × 10^−5^	5.49 × 10^−5^	4.41 × 10^−5^	**3.76 × 10^−5^**
	std	6.27868 × 10^−5^	0.0005431	4.152 × 10^−5^	4.91 × 10^−5^	4.31 × 10^−5^	**2.79 × 10^−5^**
	time	0.031770833	0.2036458	0.215625	0.041146	0.196875	0.1895833
$f_{6}$	best	0	0.9949591	0	0	0	**0**
	median	0	4.9747953	0	0	0	**0**
	worst	0.000745709	8.9546265	0	0	0	**0**
	mean	4.60805 × 10^−5^	4.8917741	0	0	0	**0**
	std	0.000172991	2.0874954	0	0	0	**0**
	time	0.0234375	0.2151042	0.2328125	0.029688	0.176563	0.1953125
$f_{7}$	best	8.88178 × 10^−16^	4.441 × 10^−15^	8.882 × 10^−16^	8.88 × 10^−16^	8.88 × 10^−16^	8.88 × 10^−16^
	median	8.88178 × 10^−16^	1.1551485	8.882 × 10^−16^	8.88 × 10^−16^	8.88 × 10^−16^	8.88 × 10^−16^
	worst	8.88178 × 10^−16^	3.4041583	8.882 × 10^−16^	8.88 × 10^−16^	8.88 × 10^−16^	8.88 × 10^−16^
	mean	8.88178 × 10^−16^	1.4393757	8.882 × 10^−16^	8.88 × 10^−16^	8.88 × 10^−16^	8.88 × 10^−16^
	std	9.86076 × 10^−32^	1.0978283	9.861 × 10^−32^	9.86 × 10^−32^	9.86 × 10^−32^	**9.86 × 10^−32^**
	time	0.026041667	0.2114583	0.2223958	0.026042	0.182292	0.1958333
$f_{8}$	best	0	0.0615239	0	0	0	**0**
	median	0	0.5017876	0	0	0	**0**
	worst	0	1.7593448	0	0	0	**0**
	mean	0	0.562363	0	0	0	**0**
	std	0	0.3632474	0	0	0	**0**
	time	0.028125	0.2192708	0.2213542	0.035417	0.198438	0.2182292
$f_{9}$	best	3.80995 × 10^−10^	4.712 × 10^−32^	4.712 × 10^−32^	5.18 × 10^−9^	3.32 × 10^−24^	**4.71 × 10^−32^**
	median	1.75313 × 10^−7^	4.712 × 10^−32^	4.712 × 10^−32^	2.1 × 10^−7^	2.24 × 10^−21^	**4.71 × 10^−32^**
	worst	6.7531 × 10^−6^	0.9328919	4.695 × 10^−30^	1.46 × 10^−5^	4.1 × 10^−19^	**4.71 × 10^−32^**
	mean	1.27274 × 10^−6^	0.0621971	2.035 × 10^−31^	1.87 × 10^−6^	2.98 × 10^−20^	**4.71 × 10^−32^**
	std	1.86588 × 10^−6^	0.1865841	8.34 × 10^−31^	3.21 × 10^−6^	7.8 × 10^−20^	**1.64 × 10^−47^**
	time	0.057291667	0.2390625	0.2578125	0.069792	0.233854	0.2317708
$f_{10}$	best	1.42853 × 10^−8^	1.35 × 10^−32^	1.35 × 10^−32^	3.3 × 10^−10^	7.06 × 10^−23^	**1.35 × 10^−32^**
	median	9.68122 × 10^−7^	1.35 × 10^−32^	1.35 × 10^−32^	1.68 × 10^−6^	4.71 × 10^−20^	**1.35 × 10^−32^**
	worst	1.27366 × 10^−5^	0.0109874	1.35 × 10^−32^	8.48 × 10^−5^	0.097371	**1.35 × 10^−32^**
	mean	2.47232 × 10^−6^	0.0032962	1.35 × 10^−32^	9.48 × 10^−6^	0.01083	**1.35 × 10^−32^**
	std	3.26171 × 10^−6^	0.005035	5.474 × 10^−48^	1.73 × 10^−5^	0.022187	**5.47 × 10^−48^**
	time	0.05625	0.2432292	0.2703125	0.076563	0.234375	0.2348958

**Table 6 biomimetics-08-00381-t006:** Comparison between algorithms for $f_{11}$–$f_{19}$.

Function		AO	MOA	AMOA	OBLAO	OBLMOA	AOBLMOA
$f_{11}$	best	0.0003132	0.0003075	0.0003075	0.000317	0.000307	**0.000307**
	median	0.0004223	0.0003075	0.0003075	0.000398	0.000307	**0.000307**
	worst	0.0006218	0.0003075	0.0012232	0.000712	0.000424	**0.000307**
	mean	0.0004356	0.0003075	0.000338	0.000438	0.000311	**0.000307**
	std	7.281 × 10^−5^	0	0.0001644	9.7 × 10^−5^	2.1 × 10^−5^	**0**
	time	0.0223958	0.1989583	0.2140625	0.025521	0.174479	0.188021
$f_{12}$	best	−1.031628	−1.031628	−1.031628	−1.03163	−1.03163	**−1.03163**
	median	−1.031486	−1.031628	−1.031628	−1.0316	−1.03163	**−1.03163**
	worst	−1.030616	−1.031628	−1.031628	−1.03106	−1.03163	**−1.03163**
	mean	−1.031424	−1.031628	−1.031628	−1.03156	−1.03163	**−1.03163**
	std	0.0002147	0	0	0.000106	0	**0**
	time	0.01875	0.1932292	0.2088542	0.021354	0.171354	0.190104
$f_{13}$	best	0.3978875	0.3978874	0.3978874	0.397887	0.397887	**0.397887**
	median	0.3979278	0.3978874	0.3978874	0.397921	0.397887	**0.397887**
	worst	0.3985984	0.3978874	0.3978874	0.398176	0.397887	**0.397887**
	mean	0.397974	0.3978874	0.3978874	0.397954	0.397887	**0.397887**
	std	0.0001446	1.11 × 10^−16^	1.11 × 10^−16^	7.31 × 10^−5^	1.11 × 10^−16^	**1.11 × 10^−16^**
	time	0.0239583	0.1916667	0.215625	0.025521	0.178125	0.183854
$f_{14}$	best	3.0000982	3	3	3.000082	3	**3**
	median	3.0085938	3	3	3.003795	3	**3**
	worst	3.0327635	3	3	3.066698	3	**3**
	mean	3.0117942	3	3	3.01013	3	**3**
	std	0.0090679	2.979 × 10^−15^	2.446 × 10^−15^	0.013312	4.53 × 10^−15^	**1.92 × 10^−15^**
	time	0.0171875	0.1942708	0.2052083	0.01875	0.166146	0.174479
$f_{15}$	best	−3.862622	−3.862782	−3.862782	−3.86252	−3.86278	**−3.86278**
	median	−3.85822	−3.862782	−3.862782	−3.86007	−3.86278	**−3.86278**
	worst	−3.852219	−3.862782	−3.862782	−3.85032	−3.08976	**−3.86278**
	mean	−3.857847	−3.862782	−3.862782	−3.85924	−3.83701	**−3.86278**
	std	0.0031974	2.665 × 10^−15^	2.665 × 10^−15^	0.002795	0.138761	**2.66 × 10^−15^**
	time	0.0244792	0.2072917	0.2140625	0.023438	0.186458	0.205208
$f_{16}$	best	−3.317657	−3.321995	−3.321995	−3.31361	−3.322	**−3.322**
	median	−3.187066	−3.321995	−3.321995	−3.27522	−3.322	**−3.322**
	worst	−2.983285	−3.203102	−3.203102	−3.11183	−3.2031	**−3.322**
	mean	−3.192639	−3.270475	−3.29029	−3.24649	−3.29425	**−3.322**
	std	0.0862405	0.0589158	0.0525765	0.057575	0.050286	**1.33 × 10^−15^**
	time	0.0260417	0.2098958	0.2130208	0.025521	0.177604	0.197917
$f_{17}$	best	−10.15318	−10.1532	−10.1532	−10.1532	−10.1532	**−10.1532**
	median	−10.15134	−10.1532	−6.589102	−10.1532	−10.1532	**−10.1532**
	worst	−10.13038	−2.630472	−5.18484	−10.1531	−10.1532	**−10.1532**
	mean	−10.14889	−6.74062	−6.957913	−10.1532	−10.1532	**−10.1532**
	std	0.0056397	3.6654023	1.6095003	3.47 × 10^−5^	1.78 × 10^−15^	**1.78 × 10^−15^**
	time	0.0260417	0.1984375	0.2125	0.028125	0.18125	0.200521
$f_{18}$	best	−10.4029	−10.40294	−10.40294	−10.4029	−10.4029	**−10.4029**
	median	−10.40193	−10.40294	−8.049912	−10.4028	−10.4029	**−10.4029**
	worst	−10.35991	−2.751934	−5.198423	−10.4026	−10.4029	**−10.4029**
	mean	−10.39792	−8.938495	−7.787828	−10.4028	−10.4029	**−10.4029**
	std	0.0088619	2.9359557	1.8213969	8.36 × 10^−5^	0	**0**
	time	0.0302083	0.1994792	0.2197917	0.03125	0.185417	0.217708
$f_{19}$	best	−10.53628	−10.53641	−10.53641	−10.5364	−10.5364	**−10.5364**
	median	−10.53504	−10.53641	−8.006939	−10.5363	−10.5364	**−10.5364**
	worst	−10.50195	−2.421734	−5.206245	−10.5356	−10.0647	**−10.5364**
	mean	−10.53173	−8.358936	−7.621057	−10.5362	−10.5212	**−10.5364**
	std	0.0074754	3.4193071	1.8240779	0.000144	0.083339	**2.57 × 10^−14^**
	time	0.040625	0.2125	0.21875	0.042708	0.198438	0.202604

**Table 7 biomimetics-08-00381-t007:** Comparison between algorithms for $f_{1}$–$f_{10}$; dimension fixed to 30.

Function		AO	MOA	AMOA	OBLAO	OBLMOA	AOBLMOA
$f_{1}$	best	8.68 × 10^−305^	2.98 × 10^−31^	1.96 × 10^−295^	0	0	**0**
	median	1.80 × 10^−289^	1.49 × 10^−27^	4.26 × 10^−253^	0	0	**0**
	worst	2.56 × 10^−200^	9.59 × 10^−24^	2.23 × 10^−218^	0	0	**0**
	mean	8.55 × 10^−202^	7.99 × 10^−25^	7.45 × 10^−220^	0	0	**0**
	std	0	2.404 × 10^−24^	0	0	0	**0**
$f_{2}$	best	1.62 × 10^−149^	1.031 × 10^−16^	1.95 × 10^−132^	0	0	**0**
	median	4.75 × 10^−145^	3.868 × 10^−15^	6.69 × 10^−117^	0	0	**0**
	worst	1.52 × 10^−105^	3.198 × 10^−11^	4.38 × 10^−102^	0	0	**0**
	mean	5.94 × 10^−107^	1.268 × 10^−12^	1.46 × 10^−103^	0	0	**0**
	std	2.76 × 10^−106^	5.725 × 10^−12^	7.87 × 10^−103^	0	0	**0**
$f_{3}$	best	1.92 × 10^−300^	4.573 × 10^−8^	5.38 × 10^−238^	0	0	**0**
	median	5.85 × 10^−287^	2.922 × 10^−7^	2.01 × 10^−198^	0	0	**0**
	worst	4.65 × 10^−198^	9.657 × 10^−7^	4.14 × 10^−182^	0	0	**0**
	mean	1.55 × 10^−199^	3.267 × 10^−7^	1.38 × 10^−183^	0	0	**0**
	std	0	2.043 × 10^−7^	0	0	0	**0**
$f_{4}$	best	1.93 × 10^−150^	0.0427667	2.23 × 10^−234^	0	0	**0**
	median	2.33 × 10^−146^	0.1220853	1.09 × 10^−194^	0	0	**0**
	worst	3.04 × 10^−107^	0.5067965	7.07 × 10^−159^	0	0	**0**
	mean	1.01 × 10^−108^	0.1604663	2.38 × 10^−160^	0	0	**0**
	std	5.46 × 10^−108^	0.1149414	0	0	0	**0**
$f_{5}$	best	**4.74 × 10^−7^**	0.0035187	3.094 × 10^−6^	1.27 × 10^−6^	1.29 × 10^−6^	2.59 × 10^−6^
	median	4.165 × 10^−5^	0.0087836	3.066 × 10^−5^	**1.5 × 10^−5^**	2.41 × 10^−5^	1.68 × 10^−5^
	worst	0.0001667	0.0158472	0.0002258	0.000146	0.000153	**0.000145**
	mean	5.532 × 10^−5^	0.0092361	4.892 × 10^−5^	3.3 × 10^−5^	4 × 10^−5^	**3.24 × 10^−5^**
	std	4.37 × 10^−5^	0.0037155	4.926 × 10^−5^	3.39 × 10^−5^	3.62 × 10^−5^	**3.1 × 10^−5^**
$f_{6}$	best	0	10.94455	0	0	0	**0**
	median	0	15.919345	0	0	0	**0**
	worst	0	25.868925	1.442 × 10-08	0	0	**0**
	mean	0	16.28416	4.929 × 10-10	0	0	**0**
	std	0	3.4416865	2.588 × 10-09	0	0	**0**
$f_{7}$	best	8.882 × 10^−16^	1.5017466	8.882 × 10^−16^	8.88 × 10^−16^	8.88 × 10^−16^	**8.88 × 10^−16^**
	median	8.882 × 10^−16^	4.424305	8.882 × 10^−16^	8.88 × 10^−16^	8.88 × 10^−16^	**8.88 × 10^−16^**
	worst	8.882 × 10^−16^	6.6919503	8.882 × 10^−16^	8.88 × 10^−16^	8.88 × 10^−16^	**8.88 × 10^−16^**
	mean	8.882 × 10^−16^	4.7257154	8.882 × 10^−16^	8.88 × 10^−16^	8.88 × 10^−16^	**8.88 × 10^−16^**
	std	9.861 × 10^−32^	1.3101158	9.861 × 10^−32^	9.86 × 10^−32^	9.86 × 10^−32^	**9.86 × 10^−32^**
$f_{8}$	best	0	0	0	0	0	**0**
	median	0	0.0098573	0	0	0	**0**
	worst	0	0.0442976	0	0	0	**0**
	mean	0	0.013043	0	0	0	**0**
	std	0	0.0125127	0	0	0	**0**
$f_{9}$	best	6.255 × 10^−10^	4.122 × 10^−28^	7.453 × 10^−11^	1.1 × 10^−9^	9.93 × 10^−7^	**1.57 × 10^−32^**
	median	7.472 × 10^−8^	1.97 × 10^−24^	2.31 × 10^−9^	2.62 × 10^−7^	4.25 × 10^−6^	**1.73 × 10^−32^**
	worst	2.872 × 10^−6^	1.7645012	6.044 × 10^−8^	1.2 × 10^−5^	3.11 × 10^−5^	**1.56 × 10^−30^**
	mean	5.104 × 10^−7^	0.2455107	6.531 × 10^−9^	9.54 × 10^−7^	6.99 × 10^−6^	**2.09 × 10^−31^**
	std	7.581 × 10^−7^	0.3905505	1.339 × 10^−8^	2.18 × 10^−6^	6.67 × 10^−6^	**4.54 × 10^−31^**
$f_{10}$	best	1.551 × 10^−9^	5.583 × 10^−31^	3.387 × 10^−10^	7.66 × 10^−8^	0.002584	**1.35 × 10^−32^**
	median	1.296 × 10^−6^	0.0988826	1.925 × 10^−8^	2.37 × 10^−6^	2.966079	**1.84 × 10^−32^**
	worst	8.757 × 10^−5^	3.866934	0.0210238	5.36 × 10^−5^	2.966171	**1.09 × 10^−29^**
	mean	1.378 × 10^−5^	0.8587114	0.0017997	1.02 × 10^−5^	2.801101	**4.02 × 10^−31^**
	std	2.075 × 10^−5^	1.2100084	0.0048546	1.44 × 10^−5^	0.584938	**1.95 × 10^−30^**

**Table 8 biomimetics-08-00381-t008:** Comparison between algorithms for $f_{1}$–$f_{10}$; dimension fixed to 50.

Function		AO	MOA	AMOA	OBLAO	OBLMOA	AOBLMOA
$f_{1}$	best	4.68 × 10^−303^	4.80 × 10^−13^	2.01 × 10^−241^	0	0	**0**
	median	1.24 × 10^−289^	3.65 × 10^−11^	5.11 × 10^−210^	0	0	**0**
	worst	2.89 × 10^−198^	1.18 × 10^−7^	2.07 × 10^−179^	0	0	**0**
	mean	9.68 × 10^−200^	4.13 × 10^−9^	6.91 × 10^−181^	0	0	**0**
	std	0	2.107 × 10^−8^	0	0	0	**0**
$f_{2}$	best	1.78 × 10^−149^	9.361 × 10^−8^	3 × 10^−118^	0	0	**0**
	median	1.77 × 10^−142^	5.794 × 10^−6^	4.83 × 10^−105^	0	0	**0**
	worst	2.07 × 10^−98^	0.0842874	1.581 × 10^−92^	0	0	**0**
	mean	7.71 × 10^−100^	0.0028957	5.567 × 10^−94^	0	0	**0**
	std	3.72 × 10^−99^	0.0151172	2.835 × 10^−93^	0	0	**0**
$f_{3}$	best	5.69 × 10^−296^	0.0386125	3.44 × 10^−222^	0	0	**0**
	median	8.65 × 10^−285^	0.1829172	1.52 × 10^−183^	0	0	**0**
	worst	1.36 × 10^−199^	0.7213506	1.72 × 10^−149^	0	0	**0**
	mean	4.53 × 10^−201^	0.2411581	5.72 × 10^−151^	0	0	**0**
	std	0	0.1736175	3.08 × 10^−150^	0	0	**0**
$f_{4}$	best	7.51 × 10^−157^	1.0774898	3.35 × 10^−259^	0	0	**0**
	median	8.13 × 10^−147^	3.8435003	4.38 × 10^−199^	0	0	**0**
	worst	1.98 × 10^−99^	6.5126095	1.32 × 10^−147^	0	0	**0**
	mean	6.64 × 10^−101^	3.8392257	4.39 × 10^−149^	0	0	**0**
	std	3.55 × 10^−100^	1.3039503	2.37 × 10^−148^	0	0	**0**
$f_{5}$	best	1.432 × 10^−6^	0.0172656	5.276 × 10^−6^	1.09 × 10^−6^	1.3 × 10^−6^	**3.63 × 10^−7^**
	median	3.38 × 10^−5^	0.0311983	4.656 × 10^−5^	2.04 × 10^−5^	2.32 × 10^−5^	**1.82 × 10^−5^**
	worst	0.0001686	0.0554084	0.0001994	0.000189	0.000193	**9.02 × 10^−5^**
	mean	5.087 × 10^−5^	0.0356282	6.119 × 10^−5^	4.01 × 10^−5^	4.14 × 10^−5^	**2.78 × 10^−5^**
	std	4.552 × 10^−5^	0.0104557	4.796 × 10^−5^	4.26 × 10^−5^	4.41 × 10^−5^	**2.45 × 10^−5^**
$f_{6}$	best	0	18.904222	0	0	0	**0**
	median	0	24.873976	0	0	0	**0**
	worst	0	41.788265	0	0	0	**0**
	mean	0	28.356325	0	0	0	**0**
	std	0	6.2280992	0	0	0	**0**
$f_{7}$	best	8.882 × 10^−16^	3.4734862	8.882 × 10^−16^	8.882 × 10^−16^	8.882 × 10^−16^	**8.882 × 10^−16^**
	median	8.882 × 10^−16^	6.9923592	8.882 × 10^−16^	8.882 × 10^−16^	8.882 × 10^−16^	**8.882 × 10^−16^**
	worst	8.882 × 10^−16^	9.2227609	8.882 × 10^−16^	8.882 × 10^−16^	8.882 × 10^−16^	**8.882 × 10^−16^**
	mean	8.882 × 10^−16^	6.9560715	8.882 × 10^−16^	8.882 × 10^−16^	8.882 × 10^−16^	**8.882 × 10^−16^**
	std	9.861 × 10^−32^	1.305546	9.861 × 10^−32^	9.86 × 10^−32^	9.86 × 10^−32^	**9.86 × 10^−32^**
$f_{8}$	best	0	2.946 × 10-11	0	0	0	**0**
	median	0	6.311 × 10-10	0	0	0	**0**
	worst	0	0.0270517	0	0	0	**0**
	mean	0	0.0052508	0	0	0	**0**
	std	0	0.0079486	0	0	0	**0**
$f_{9}$	best	5.258 × 10^−9^	3.272 × 10^−12^	1.196 × 10^−7^	3.25 × 10^−9^	0.000216	**3.9 × 10^−23^**
	median	1.828 × 10^−7^	0.0622014	5.664 × 10^−7^	1.58 × 10^−7^	0.000472	**6.89 × 10^−19^**
	worst	1.855 × 10^−6^	1.372991	9.859 × 10^−6^	6.87 × 10^−6^	0.0018	**2.08 × 10^−16^**
	mean	3.418 × 10^−7^	0.2886864	1.568 × 10^−6^	1.11 × 10^−6^	0.000619	**2.23 × 10^−17^**
	std	4.312 × 10^−7^	0.3724752	2.328 × 10^−6^	1.75 × 10^−6^	0.000385	**5.15 × 10^−17^**
$f_{10}$	best	4.392 × 10^−8^	0.0112602	1.616 × 10^−6^	1.58 × 10^−8^	4.943478	**1.97 × 10^−19^**
	median	1.771 × 10^−6^	3.0772429	1.662 × 10^−5^	2.4 × 10^−6^	4.944099	**1.98 × 10^−17^**
	worst	3.427 × 10^−5^	37.545497	0.0212338	7.5 × 10^−5^	4.946246	**1.39 × 10^−15^**
	mean	4.587 × 10^−6^	8.6367625	0.0028991	1.06 × 10^−5^	4.944325	**1.92 × 10^−16^**
	std	0.0001376	259.10287	0.086973	0.000319	148.3298	**5.76 × 10^−15^**

**Table 9 biomimetics-08-00381-t009:** Comparison between algorithms for $f_{1}$–$f_{10}$; dimension fixed to 100.

Function		AO	MOA	AMOA	OBLAO	OBLMOA	AOBLMOA
$f_{1}$	best	1.60 × 10^−306^	1.56 × 10^−7^	7.15 × 10^−246^	0	0	**0**
	median	1.12 × 10^−291^	2.28 × 10^−7^	2.1 × 10^−196^	0	0	**0**
	worst	2.85 × 10^−191^	4.03 × 10^−7^	2.99 × 10^−150^	0	0	**0**
	mean	9.51 × 10^−193^	2.46 × 10^−7^	9.98 × 10^−152^	0	0	**0**
	std	0	6.112 × 10^−8^	5.37 × 10^−151^	0	0	**0**
$f_{2}$	best	3.71 × 10^−149^	0.00655	4.42 × 10^−114^	0	0	**0**
	median	7.33 × 10^−144^	0.0232734	4.278 × 10^−86^	0	0	**0**
	worst	4.1 × 10^−101^	0.7581851	4.327 × 10^−73^	0	0	**0**
	mean	1.64 × 10^−102^	0.0683458	1.47 × 10^−74^	0	0	**0**
	std	7.38 × 10^−102^	0.1355721	7.763 × 10^−74^	0	0	**0**
$f_{3}$	best	2.39 × 10^−298^	104.39766	1.28 × 10^−190^	0	0	**0**
	median	2.78 × 10^−282^	154.47363	2.64 × 10^−167^	0	0	**0**
	worst	8.55 × 10^−197^	486.07907	1.09 × 10^−117^	0	0	**0**
	mean	2.85 × 10^−198^	180.48085	3.63 × 10^−119^	0	0	**0**
	std	0	76.947176	1.96 × 10^−118^	0	0	**0**
$f_{4}$	best	1.68 × 10^−153^	8.3007886	7.35 × 10^−267^	0	0	**0**
	median	3.91 × 10^−146^	11.790905	5.49 × 10^−220^	0	0	**0**
	worst	1.332 × 10^−98^	15.014142	8.14 × 10^−146^	0	0	**0**
	mean	6.4 × 10^−100^	11.827796	2.71 × 10^−147^	0	0	**0**
	std	2.58 × 10^−99^	1.5147406	1.46 × 10^−146^	0	0	**0**
$f_{5}$	best	4.286 × 10^−7^	0.1149912	1.235 × 10^−6^	2.3 × 10^−6^	5.28 × 10^−6^	**2.20 × 10^−7^**
	median	2.655 × 10^−5^	0.1595987	3.08 × 10^−5^	1.53 × 10^−5^	4.28 × 10^−5^	**2.18 × 10^−5^**
	worst	0.0001672	0.2613939	0.0001912	0.00015	0.000225	**0.000102**
	mean	4.542 × 10^−5^	0.168212	4.66 × 10^−5^	3.33 × 10^−5^	6.83 × 10^−5^	**3.04 × 10^−5^**
	std	4.359 × 10^−5^	0.0317316	4.736 × 10^−5^	3.05 × 10^−5^	5.19 × 10^−5^	**2.73 × 10^−5^**
$f_{6}$	best	0	39.798413	0	0	0	**0**
	median	0	57.707615	0	0	0	**0**
	worst	0	77.606781	0	0	0	**0**
	mean	0	58.702617	0	0	0	**0**
	std	0	8.4190155	0	0	0	**0**
$f_{7}$	best	8.882 × 10^−16^	5.6149166	8.882 × 10^−16^	8.882 × 10^−16^	8.882 × 10^−16^	**8.882 × 10^−16^**
	median	8.882 × 10^−16^	8.3966259	8.882 × 10^−16^	8.882 × 10^−16^	8.882 × 10^−16^	**8.882 × 10^−16^**
	worst	8.882 × 10^−16^	10.024012	8.882 × 10^−16^	8.882 × 10^−16^	8.882 × 10^−16^	**8.882 × 10^−16^**
	mean	8.882 × 10^−16^	8.3048826	8.882 × 10^−16^	8.882 × 10^−16^	8.882 × 10^−16^	**8.882 × 10^−16^**
	std	9.861 × 10^−32^	1.0122703	9.861 × 10^−32^	9.861 × 10^−32^	9.861 × 10^−32^	**9.861 × 10^−32^**
$f_{8}$	best	0	0.0011253	0	0	0	**0**
	median	0	0.010309	0	0	0	**0**
	worst	0	0.042127	0	0	0	**0**
	mean	0	0.0151088	0	0	0	**0**
	std	0	0.0104558	0	0	0	**0**
$f_{9}$	best	3.442 × 10^−10^	0.4899941	8.979 × 10^−6^	1.38 × 10^−9^	0.006845	**3.28 × 10^−13^**
	median	8.43 × 10^−8^	1.4336532	6.114 × 10^−5^	2.17 × 10^−7^	0.011347	**4.08 × 10^−11^**
	worst	5.784 × 10^−6^	4.9763026	0.0013151	3.84 × 10^−6^	0.032085	**1.56 × 10^−9^**
	mean	5.383 × 10^−7^	1.8083967	0.0002638	6.05 × 10^−7^	0.013713	**2.94 × 10^−10^**
	std	1.153 × 10^−6^	0.9399669	0.0003789	8.51 × 10^−7^	0.005306	**3.97 × 10^−10^**
$f_{10}$	best	2.305 × 10^−8^	66.782646	0.0001677	1.98 × 10^−7^	9.895722	**7.46 × 10^−13^**
	median	6.757 × 10^−6^	95.956055	0.003001	5.48 × 10^−6^	9.900694	**1.53 × 10^−9^**
	worst	0.0002711	149.9168	0.1092565	0.000202	9.912946	**1.58 × 10^−7^**
	mean	2.877 × 10^−5^	99.021078	0.0116608	2.27 × 10^−5^	9.90212	**1.96 × 10^−8^**
	std	5.867 × 10^−5^	20.666644	0.0206942	4.24 × 10^−5^	0.003752	**3.75 × 10^−8^**

**Table 10 biomimetics-08-00381-t010:** Wilcoxon rank sum test under benchmark function.

AOBLMOA vs.	Dim	AO	MOA	AMOA	OBLAO	OBLMOA
$f_{1}$	10	1.73 × 10−6	1.73 × 10−6	1	1	1
	30	1.73 × 10−6	1.73 × 10−6	1.73 × 10−6	1	1
	50	1.73 × 10−6	1.73 × 10−6	1.73 × 10−6	1	1
	100	1.73 × 10−6	1.73 × 10−6	1.73 × 10−6	1	1
$f_{2}$	10	1.73 × 10−6	1.73 × 10−6	1.73 × 10−6	1	1
	30	1.73 × 10−6	1.73 × 10−6	1.73 × 10−6	1	1
	50	1.73 × 10−6	1.73 × 10−6	1.73 × 10−6	1	1
	100	1.73 × 10−6	1.73 × 10−6	1.73 × 10−6	1	1
$f_{3}$	10	1.73 × 10−6	1.73 × 10−6	1.73 × 10−6	1	1
	30	1.73 × 10−6	1.73 × 10−6	1.73 × 10−6	1	1
	50	1.73 × 10−6	1.73 × 10−6	1.73 × 10−6	1	1
	100	1.73 × 10−6	1.73 × 10−6	1.73 × 10−6	1	1
$f_{4}$	10	1.73 × 10−6	1.73 × 10−6	1.73 × 10−6	1	1
	30	1.73 × 10−6	1.73 × 10−6	1.73 × 10−6	1	1
	50	1.73 × 10−6	1.73 × 10−6	1.73 × 10−6	1	1
	100	1.73 × 10−6	1.73 × 10−6	1.73 × 10−6	1	1
$f_{5}$	10	0.280214	1.73 × 10−6	0.308615	0.22888	0.765519
	30	0.033269	1.73 × 10−6	0.009842	0.349333	0.271155
	50	0.033269	1.73 × 10−6	0.009842	0.349333	0.271155
	100	0.42843	1.73 × 10−6	0.338843	0.557743	0.002585
$f_{6}$	10	0.5	1.73 × 10−6	1	1	1
	30	1	1.73 × 10−6	1	1	1
	50	1	1.73 × 10−6	1	1	1
	100	1	1.73 × 10−6	1	1	1
$f_{7}$	10	1	1.73 × 10−6	1	1	1
	30	1	1.73 × 10−6	1	1	1
	50	1	1.73 × 10−6	1	1	1
	100	1	1.73 × 10−6	1	1	1
$f_{8}$	10	1	1.73 × 10−6	1	1	1
	30	1	1.73 × 10−6	1	1	1
	50	1	1.73 × 10−6	1	1	1
	100	1	1.73 × 10−6	1	1	1
$f_{9}$	10	1.73 × 10−6	0.0625	0.000122	1.73 × 10−6	1.73 × 10−6
	30	1.73 × 10−6	1.73 × 10−6	1.73 × 10−6	1.73 × 10−6	1.73 × 10−6
	50	1.73 × 10−6	1.73 × 10−6	1.73 × 10−6	1.73 × 10−6	1.73 × 10−6
	100	1.73 × 10−6	1.73 × 10−6	1.73 × 10−6	1.73 × 10−6	1.73 × 10−6
$f_{10}$	10	1.73 × 10−6	0.003906	1	1.73 × 10−6	1.73 × 10−6
	30	1.73 × 10−6	1.73 × 10−6	1.73 × 10−6	1.73 × 10−6	1.73 × 10−6
	50	1.73 × 10−6	1.73 × 10−6	1.73 × 10−6	1.73 × 10−6	1.73 × 10−6
	100	1.73 × 10−6	1.73 × 10−6	1.73 × 10−6	1.73 × 10−6	1.73 × 10−6
$f_{11}$	4	1.73 × 10−6	1	1	1.73 × 10−6	1
$f_{12}$	2	1.73 × 10−6	1	1	1.73 × 10−6	1
$f_{13}$	2	0.000453	0.25	0.25	0.000241	0.25
$f_{14}$	2	1.73 × 10−6	1	1	1.73 × 10−6	1
$f_{15}$	3	1.73 × 10−6	1	1	1.73 × 10−6	1
$f_{16}$	6	1.73 × 10−6	0.000244	0.007813	1.73 × 10−6	0.015625
$f_{17}$	4	1.73 × 10−6	0.000122	3.79 × 10−6	1.73 × 10−6	1
$f_{18}$	4	1.73 × 10−6	0.03125	3.79 × 10−6	1.73 × 10−6	1
$f_{19}$	4	1.73 × 10−6	0.003906	3.79 × 10−6	1.73 × 10−6	1
+/−/=	-	0/35/14	0/43/6	0/28/21	0/17/32	0/10/39

**Table 11 biomimetics-08-00381-t011:** Friedman rank sum test under benchmark function.

Function	Dim	AO	MOA	AMOA	OBLAO	OBLMOA	AOBLMOA
$f_{1}$	10	5	6	1	1	1	1
	30	5	6	4	1	1	1
	50	4	6	5	1	1	1
	100	4	6	5	1	1	1
$f_{2}$	10	5	6	4	1	1	1
	30	4	6	5	1	1	1
	50	4	6	5	1	1	1
	100	4	6	5	1	1	1
$f_{3}$	10	5	6	4	1	1	1
	30	4	6	5	1	1	1
	50	4	6	5	1	1	1
	100	4	6	5	1	1	1
$f_{4}$	10	5	6	4	1	1	1
	30	5	6	4	1	1	1
	50	5	6	4	1	1	1
	100	5	6	4	1	1	1
$f_{5}$	10	5	6	3	4	2	1
	30	5	6	4	2	3	1
	50	4	6	5	2	3	1
	100	3	6	4	2	5	1
$f_{6}$	10	5	6	1	1	1	1
	30	1	6	5	1	1	1
	50	1	6	1	1	1	1
	100	1	6	1	1	1	1
$f_{7}$	10	1	6	1	1	1	1
	30	1	6	1	1	1	1
	50	1	6	1	1	1	1
	100	1	6	1	1	1	1
$f_{8}$	10	1	6	1	1	1	1
	30	1	6	1	1	1	1
	50	1	6	1	1	1	1
	100	1	6	1	1	1	1
$f_{9}$	10	4	6	2	5	3	1
	30	3	6	2	4	5	1
	50	2	6	4	3	5	1
	100	2	6	4	3	5	1
$f_{10}$	10	3	5	1	4	6	1
	30	3	5	4	2	6	1
	50	2	6	4	3	5	1
	100	3	6	4	2	5	1
$f_{11}$	4	5	1	4	6	3	1
$f_{12}$	2	6	1	1	5	1	1
$f_{13}$	2	6	1	1	5	1	1
$f_{14}$	2	6	1	1	5	4	1
$f_{15}$	3	5	1	1	4	6	1
$f_{16}$	6	6	4	3	5	2	1
$f_{17}$	4	4	6	5	3	1	1
$f_{18}$	4	4	5	6	3	1	1
$f_{19}$	4	3	5	6	2	4	1
Friedman rank	-	3.510204	5.367347	3.142857	2.081633	2.122449	1
Final rank	-	5	6	4	2	3	1

**Table 12 biomimetics-08-00381-t012:** Summary of the CEC2017 test functions.

Type	Func No.	Details	Fi
Unimodal Functions	$f_{1}$	Shifted and Rotated Bent Cigar Function	100
	$f_{2}$	Shifted and Rotated Sum Diff Pow Function	200
	$f_{3}$	Shifted and Rotated Zakharov Function	300
Simple Multimodal Functions	$f_{4}$	Shifted and Rotated Rosenbrock’s Function	400
	$f_{5}$	Shifted and Rotated Rastrigin’s Function	500
	$f_{6}$	Shifted and Rotated Expanded Scaffer’s F6 Function	600
	$f_{7}$	Shifted and Rotated Lunacek Bi_Rastrigin Function	700
	$f_{8}$	Shifted and Rotated Non-Continuous Rastrigin’s Function	800
	$f_{9}$	Shifted and Rotated Lévy Function	900
	$f_{10}$	Shifted and Rotated Schwefel’s Function	1000
Hybrid Functions	$f_{11}$	Hybrid Function 1 (N = 3)	1100
	$f_{12}$	Hybrid Function 2 (N = 3)	1200
	$f_{13}$	Hybrid Function 3 (N = 3)	1300
	$f_{14}$	Hybrid Function 4 (N = 4)	1400
	$f_{15}$	Hybrid Function 5 (N = 4)	1500
	$f_{16}$	Hybrid Function 6 (N = 4)	1600
	$f_{17}$	Hybrid Function 6 (N = 5)	1700
	$f_{18}$	Hybrid Function 6 (N = 5)	1800
	$f_{19}$	Hybrid Function 6 (N = 5)	1900
	$f_{20}$	Hybrid Function 6 (N = 6)	2000
Composition Functions	$f_{21}$	Composition Function 1 (N = 3)	2100
	$f_{22}$	Composition Function 2 (N = 3)	2200
	$f_{23}$	Composition Function 3 (N = 4)	2300
	$f_{24}$	Composition Function 4 (N = 4)	2400
	$f_{25}$	Composition Function 5 (N = 5)	2500
	$f_{26}$	Composition Function 6 (N = 5)	2600
	$f_{27}$	Composition Function 7 (N = 6)	2700
	$f_{28}$	Composition Function 8 (N = 6)	2800
	$f_{29}$	Composition Function 9 (N = 3)	2900
	$f_{30}$	Composition Function 10 (N = 3)	3000

**Table 13 biomimetics-08-00381-t013:** Comparison of experimental results of CEC2017 test functions.

No.	Measure	AOBLMOA	MOA	AO	LGCMFO	RSA	ESSA	SGOA	COLMA
$f_{1}$	Ave	109	2721	296	9230	2470	3019	1.09 × 10^5^	2380
	STD	11	2492	275	7920	265	2362	73,218	1060
	Rank	1	5	2	7	4	6	8	3
$f_{2}$	Ave	200	200	-	2.67 × 10^13^	-	1.22 × 10^10^	6.61 × 10^10^	200
	STD	0	0	-	3.84 × 10^13^	-	5.99 × 10^10^	3.62 × 10^11^	0
	Rank	1	2	8	6	8	4	5	3
$f_{3}$	Ave	300	28,546	1142	13,500	1510	4281	303	422
	STD	0	19,842	235	3780	25	2727	1	320
	Rank	1	8	4	7	5	6	2	3
$f_{4}$	Ave	400	413	406	500	404	520	478	402
	STD	0	20	9	22	8	34	13	43
	Rank	1	5	4	7	3	8	6	2
$f_{5}$	Ave	529	614	511	628	513	600	611	540
	STD	8	22	72	28	24	24	31	21
	Rank	3	7	1	8	2	5	6	4
$f_{6}$	Ave	600	635	624	610	600	600	602	600
	STD	7	5	14	8	1	0	2	0
	Rank	1	8	7	6	1	3	5	4
$f_{7}$	Ave	777	904	715	865	713	855	783	705
	STD	23	50	2	42	4	36	17	32
	Rank	4	8	3	7	2	6	5	1
$f_{8}$	Ave	825	882	820	921	809	884	899	853
	STD	8	16	7	27	8	18	19	14
	Rank	3	5	2	8	1	6	7	4
$f_{9}$	Ave	1063	2188	900	2690	910	1737	1448	1280
	STD	103	1192	0	854	20	571	1440	979
	Rank	3	7	1	8	2	6	5	4
$f_{10}$	Ave	1968	4604	1706	4920	1410	4094	4389	4450
	STD	191	535	362	608	35	634	456	407
	Rank	3	7	2	8	1	4	5	6
$f_{11}$	Ave	1139	1230	1125	1240	1110	1248	1205	1177
	STD	16	40	23	67	11	52	36	20
	Rank	3	6	2	7	1	8	5	4
$f_{12}$	Ave	3855	12,293	10,031	1.18 × 10^6^	1.52 × 10^4^	3.09 × 10^6^	1.81 × 10^6^	4.84 × 10^5^
	STD	2176	7029	232	1.23 × 10^6^	2.68 × 10^3^	1.58 × 10^6^	9.33 × 10^5^	6.21 × 10^5^
	Rank	1	3	2	6	4	8	7	5
$f_{13}$	Ave	1564	11,592	8019	3.48 × 10^5^	6820	17,963	1.03 × 10^5^	7340
	STD	131	8175	5623	1.13 × 10^6^	4260	12,592	45,182	6320
	Rank	1	5	4	8	2	6	7	3
$f_{14}$	Ave	1442	4426	1449	39,100	1450	15,949	1851	2680
	STD	3346	5846	54	45,000	22	14,048	214	2260
	Rank	1	6	2	8	3	7	4	5
$f_{15}$	Ave	1681	2527	1710	7410	1580	5087	40,634	3780
	STD	118	555	276	6450	128	4047	22,098	1740
	Rank	2	4	3	7	1	6	8	5
$f_{16}$	Ave	1781	2627	1624	2590	1730	2109	2077	2060
	STD	118	263	40	279	120	213	146	202
	Rank	3	8	1	7	2	6	5	4
$f_{17}$	Ave	1772	2390	1742	2120	1730	1929	1974	1754
	STD	30	227	29	201	35	101	113	19
	Rank	4	8	2	7	1	5	6	3
$f_{18}$	Ave	1877	94,944	8712	2.18 × 10^5^	7440	1.67 × 10^5^	37,190	3120
	STD	46	1.39 × 10^5^	3251	1.80 × 10^5^	4520	1.45 × 10^5^	16,669	1170
	Rank	1	6	4	8	3	7	5	2
$f_{19}$	Ave	1933	4897	1944	5200	1950	5731	15,554	5150
	STD	25	3585	30	2980	55	3910	8155	3440
	Rank	1	4	2	6	3	7	8	5
$f_{20}$	Ave	2130	2457	2018	2360	2020	2316	2309	2343
	STD	76	154	21	179	25	108	160	99
	Rank	3	8	1	7	2	5	4	6
$f_{21}$	Ave	2296	2406	2205	2410	2230	2395	2415	2132
	STD	64	32	40	29	44	20	28	23
	Rank	4	6	2	7	3	5	8	1
$f_{22}$	Ave	2300	2300	2305	2300	2280	2301	5333	2300
	STD	5	2168	22	1	13	3	1487	906
	Rank	2	4	7	2	1	6	8	4
$f_{23}$	Ave	2635	3031	2620	2760	2610	2790	2752	2606
	STD	20	66	12	32	4	36	31	55
	Rank	4	8	3	6	2	7	5	1
$f_{24}$	Ave	2760	3208	2686	2920	2620	3046	2977	2594
	STD	54	69	16	29	80	54	47	66
	Rank	4	8	3	5	2	7	6	1
$f_{25}$	Ave	2918	2931	2919	2890	2920	2925	2888	2809
	STD	23	18	19	15	13	25	4	19
	Rank	4	8	5	3	6	7	2	1
$f_{26}$	Ave	3251	6565	3006	3640	3110	4947	3732	2884
	STD	451	1314	145	1280	289	1318	1069	187
	Rank	4	8	2	5	3	7	6	1
$f_{27}$	Ave	3101	3458	3090	3300	3110	3260	3200	3082
	STD	8	98	8	31	21	24	0	58
	Rank	3	8	2	7	4	6	5	1
$f_{28}$	Ave	3100	3125	3211	3230	2300	3273	3300	3169
	STD	148	52	47	22	124	25	0	63
	Rank	2	3	5	6	1	7	8	4
$f_{29}$	Ave	3663	3.74 × 10^7^	3190	3870	3210	3713	3505	8.70 × 10^6^
	STD	156	4.33 × 10^7^	29	266	57	144	124	6.98 × 10^5^
	Rank	4	8	1	6	2	5	3	7
$f_{30}$	Ave	3724	7507	290,140	33,900	2.96 × 10^5^	36,669	1.96 × 10^5^	7720
	STD	151	796	52,314	51,600	21,400	26,129	1.44 × 10^5^	1050
	Rank	1	2	7	4	8	5	6	3
	Friedman	2.433333333	6.1	3.133333333	6.466666667	2.766666667	6.033333333	5.666666667	3.333333333
	Final	1	7	3	8	2	6	5	4

**Table 14 biomimetics-08-00381-t014:** Comparison for the WMSR problem.

Algorithm	AOBLMOA	SASS	COLSHADE	sCMAgES
Ave	2994.424	2994.424	2994.424	2994.424
STD	4.54747 × 10^−13^	4.54747 × 10^−13^	4.54747 × 10^−13^	2.45564 × 10^−12^
Median	2994.424	2994.424	2994.424	2994.424
Min	2994.424	2994.424	2994.424	2994.424
Max	2994.424	2994.424	2994.424	2994.424

**Table 15 biomimetics-08-00381-t015:** Comparison for the ODIRS problem.

Algorithm	AOBLMOA	SASS	COLSHADE	sCMAgES
Ave	0.032213	0.032213	0.032213	0.036419
STD	0.000982459	1.38778 × 10^−17^	1.38778 × 10^−17^	0.001726802
Median	0.032213	0.032213	0.032213	0.036266
Min	0.032213	0.032213	0.032213	0.03355
Max	0.035048	0.032213	0.032213	0.03993

**Table 16 biomimetics-08-00381-t016:** Comparison for the TCSD1 problem.

Algorithm	AOBLMOA	SASS	COLSHADE	sCMAgES
Ave	0.012665	0.012665	0.012665	0.012668
STD	8.562 × 10^−8^	0	1.06254 × 10^−7^	4.54201 × 10^−6^
Median	0.012665	0.012665	0.012665	0.012666
Min	0.012665	0.012665	0.012665	0.012665
Max	0.012665	0.012665	0.012666	0.012669

**Table 17 biomimetics-08-00381-t017:** Comparison for the MDCBDP problem.

Algorithm	AOBLMOA	SASS	COLSHADE	sCMAgES
Ave	0.235242458	0.23524246	0.23524246	0.235242458
STD	5.55112 × 10^−17^	2.77556 × 10^−17^	2.77556 × 10^−17^	1.11022 × 10^−16^
Median	0.235242458	0.23524246	0.23524246	0.235242458
Min	0.235242458	0.23524246	0.23524246	0.235242458
Max	0.235242458	0.23524246	0.23524246	0.235242458

**Table 18 biomimetics-08-00381-t018:** Comparison for the PGTDO problem.

Algorithm	AOBLMOA	SASS	COLSHADE	sCMAgES
Ave	0.600480484	1.001524	0.541026	0.530809
STD	0.114142329	0.710533	0.042573	0.004261
Median	0.537058824	0.645573	0.53	0.53
Min	0.5271875	0.525768	0.525768	0.525967
Max	0.8795931	3.521656	0.746667	0.543846

**Table 19 biomimetics-08-00381-t019:** Comparison for the HTBDP problem.

Algorithm	AOBLMOA	SASS	COLSHADE	sCMAgES
Ave	1775.864458	1616.1201	1639.037352	3022.135455
STD	30.92850736	0.000923472	100.7282129	387.5627403
Median	1771.961306	1616.1198	1616.1198	3174.538873
Min	1708.852265	1616.1198	1616.1198	2284.476299
Max	1821.920966	1616.1234	2129.1452	3530.085832

**Table 20 biomimetics-08-00381-t020:** Comparison for the FGBP problem.

Algorithm	AOBLMOA	SASS	COLSHADE	sCMAgES
Ave	50.70603013	38.51409836	36.61097536	53.7101309
STD	15.04272341	2.072094103	1.367708578	17.51522759
Median	45.90641099	38.129703	36.249291	48.61282506
Min	35.36066244	36.250401	35.359232	36.24853345
Max	82.10915894	45.506407	40.931153	120.3173533

**Table 21 biomimetics-08-00381-t021:** Comparison for the GTCD problem.

Algorithm	AOBLMOA	SASS	COLSHADE	sCMAgES
Ave	2,964,895.4	2,964,895.4	2,964,895.4	2,964,912.352
STD	4.65661 × 10^−10^	4.65661 × 10^−10^	4.65661 × 10^−10^	34.1029386
Median	2,964,895.4	2,964,895.4	2,964,895.4	2,964,898.734
Min	2,964,895.4	2,964,895.4	2,964,895.4	2,964,895.42
Max	2,964,895.4	2,964,895.4	2,964,895.4	2,965,049.472

**Table 22 biomimetics-08-00381-t022:** Comparison for the TCSD2 problem.

Algorithm	AOBLMOA	SASS	COLSHADE	sCMAgES
Ave	2.683119896	2.6585592	2.66183396	4.236906962
STD	0.020053752	4.65661 × 10^−10^	0.011105251	1.035279415
Median	2.699493709	2.6585592	2.6585592	3.878629281
Min	2.658559166	2.6585592	2.6585592	2.852343312
Max	2.69949375	2.6585592	2.6994937	6.739677286

**Table 23 biomimetics-08-00381-t023:** Comparison for the TO problem.

Algorithm	AOBLMOA	SASS	COLSHADE	sCMAgES
Ave	2.6393465	2.6393465	2.6393465	2.6393465
STD	0	4.44089 × 10^−16^	4.44089 × 10^−16^	1.62806 × 10^−15^
Median	2.6393465	2.6393465	2.6393465	2.6393465
Min	2.6393465	2.6393465	2.6393465	2.6393465
Max	2.6393465	2.6393465	2.6393465	2.6393465

## Data Availability

Data is contained within the article. The data presented in this study are available insert article.

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
