# Peer review of "AOBLMOA: A Hybrid Biomimetic Optimization Algorithm for Numerical Optimization and Engineering Design Problems"

_biomimetics, 2023, doi:10.3390/biomimetics8040381_

Round 1

Reviewer 1 Report

The paper hybridizes mayfly optimization algorithm with the 2

aquila optimizer and opposition-based learning for numerical 3

optimization and real-world engineering design problems .

Please clear the following issues:

1. why the author choose mayfly optimization as the basis to do hybridzation.

2. the motivation is unclear

3. since the author provide a summary table of popular MHs (TABLE 1), It would be better

the author gives a brief description of algorithm characteristic for each, rather than just the inspiration/name

4. there are many parameters in the designed algorithm,  is there parameter tuning process? how you choose the parameters?

5. In Fig.1, you mentioned that ' From the trajectory, it can be seen that 459

AOBLMOA has a higher oscillation frequency in the early iteration process, ' while, in the figure, it seems, 

the population toward the same direction.

the average fitness tends to zero at around '0' iteration, that quite strange, --as the conversgence curve decrease till 200 iteration.

and what is the y-axis labels for convergence curve. 

6. Fig.2, the lengend should follows the figure, it is hard to distinguish which line represents the proposed algorithms. 

7, How do you determine the population size and iteration numbers?

8. I do not think the complexity analysis could reflect the time consumption for the hybrid algorithm. maybe the time records could be shown as extra information for readers.

Reviewer 2 Report

Abbreviations and acronyms should be avoided in the abstract.

Start the abstract with a general topic, followed by the problem you are addressing, then introduce the proposed method (only a high-level description) and findings.

For some names of the algorithms, first letter capitalization is used, sometimes in the paper the algorithm is written with lower cases, I suggest being consistent with capitalization throughout the manuscript, the same applies to section titles.

For example line 44 and line 58:

algorithms include particle swarm optimization (PSO) [3], salp search algorithm (SSA) [4], 44 

The Mayfly Optimization Algorithm (MOA) [23], proposed by Konstantinos Zervou- 58

Please consider separating the introduction and literature review, in the introduction, introduce the topic and create a separate section for the literature review.

For the equations, use display math mode, and for referencing equations in the text, use the following format, with brackets "Eq. (i)".

Correct the presentation of pseudo-code with the appropriate format.

Consider adding post-hoc procedures, as a suitable complement for the Friedman-related tests. 

Moreover, please consider addressing the following questions: What are the limitations of your method (algorithm)?

There are minor grammatical errors that should be corrected in the manuscript.

Reviewer 3 Report

In this paper, a hybrid mayfly optimization algorithm with the 2 aquila optimizer and opposition-based learning for numerical 3 optimization and real-world engineering design problems was proposed. The writing is easy to read and the work sounds interesting at the beginning (title, abstract, and introduction). But I can’t find the influence of complex communication, which is emphasized in the title and also the part I am interested in, in the model, the algorithm, and the experiments. Here are the things I suggest for revision:

1.      I suggest revising the English throughout the entire manuscript as it contains several grammatical mistakes.

2.      There are too many words in the title. The title should reflect both the content of the paper and be concise and comprehensive.

3.      In the abstract, it was not explained why this algorithm was proposed and what problem was it aimed at solving?

4.      In the introduction, there is a lack of overview and analysis of mayfly optimization algorithm.

5.      How efficient is the proposed algorithm? This paper does not theoretically analyze its computational complexity.

6.      How about the performance of the proposed algorithm for the high dimensional functions?

7.      How is the performance of the proposed algorithm compared with the recently popular improved mayfly optimization algorithm? There are too few algorithm evaluation indexes used in the experiment.

8.      The layout of the icons is neat and the format is incorrect. The legend is missing from the figure.

9.      From the experimental results, the performance of the proposed algorithm is not very obvious.

10.  In the paper, the proposed algorithm combines several methods. Which method has played a role in improving algorithm performance? The effectiveness of each method has not been demonstrated.

11.  The improvement methods of mayfly optimization algorithm in this paper have been common and not innovative enough.

Reviewer 4 Report

I recommend the acceptance of the manuscript after minor revision.

Reviewer 5 Report

1. The title is too long. Some parts of the titke are not informative, e.g. "numerical optimization" or "real-world engineering design problems" (in the end, all optimizers are designed for solving real-world problems).

2. The 1st sentence of the Abstract is just a copy of the title.

3. From the Abstract, it is not clear, for which kind of optimization problems the algorithm is designed: continuous/discrete, constrained/unconstrained, convex/con-convex, global/local optimization, large-scale/not large scale. In the end of the abstract, the authors mention constrained problems and engineering design problems. Are these endineering design problems constrained or unconstrained? The Abstract is not informative enough.

4. In the Abstract, the authors use specific terms which are not usual even for a specialist in optimization who is not damiliar with this branch of algorithms ("high soar", "verical stoop").

5. "Metaheuristic algorithms (MHs) can generally be divided into four categories: swarm 40 intelligence algorithms (SIAs), evolutionary algorithms (EAs), physics-based algorithms 41 (PhAs), and human-based algorithms (HAs)." Actually, this is a classification of the evolutionary (population-based) metaheuristic algorithms for global optimization. However, in general, the metaheuristic algorithm include more classes (e.g. greedy approaches). Thas, the Title of Table 1 must be more specific.

6. The authors mention the OBL for the first time in the end of the Abstract, without any description what it is. 

7. Th Introduction contaions a classification of the population-based meaheuristic approaches (which is incomplete and not too systematic). The components of the proposed algorithm must be described in the Introduction with more details. Moreover, it is absolute unclear why the authors choosed such a combination of algorithmic approaches for their new algorithm.

8. A sentence in line 144 starts with nothing.

9. No punctuation marks afer the euations. 

10. Symbol "*" is used for multiplication which is incorrect.

11. Incorrect indents throughout the text after equations.

12. why "dim" in (13) is not italicized? Isn't it the same as d in line 107?

13. Section 3.4 must be re-written. The formulas are not italicized, the expressions like O(MOA) are incorrect (the argument of O is not an algorithm). The presentation in this Section is chaotic.

14. It is very strange that the pseufo-code of the algorithm is given after its complexity analysis. Section 3.5 contains no test.

15. Are functions F1-F19 in Tables 3-6 the same as f_1-f_19?

16. The authors tried to show that their algorithmic combination outperforms its components. However, it is not clear, that are the measures in Tables (achieved objective function values?). It is not clear, dis all the algorithm consume the same computational resources or not (spent time and the number of the objective function estimations are not indicated).

17. For the test problems, the authors compare the results with the components of the proposed algorithm. For the other problemsm the authors compare the results with the other algorithms such as SHADE. Why? Probably, the separate components of the proposed algorithm are even better for these problems?

18. For all the algorithms used for the comparison, all the parameter values must be clearly stated and substantiated. Probably, the known algorithms work poorly because of the incorrect hyperparameter settings?

Formally, English is correct. However, the style must be improved.

Round 2

Reviewer 1 Report

the author should always give credit to the algorithm that first published it. 

which means, you should cite the first paper that proposed it, then add some new references introducing the improved ones. 

Response: extra sensitivity analysis on parameter might be informative for readers to use your algorithms.

Response: extra sensitivity analysis on parameter might be informative for readers to use your algorithms.

Response: the problem is not the legend itself, it is how to use the legend to read your figures. Some lines overlap, it is better use marker to distinguish your algorithms. So the readers know how your algorithm perform.

Response: the reviewer might ask about the ‘characteristic’, not about the detail.

Dear author, the purpose of reviewr’s question is to find the unclear part of your paper, and wish you to improve it for readers, not just for reviewers. Your answer should be informative for other readers, not just answer for nothing. 

Reviewer 3 Report

  • This paper has been carefully revised according to the comments of the reviewer, but there are still many mistakes in English grammar, so it is suggested to revise it.

Reviewer 5 Report

The authors improved the paper significantly.

Nevertheless, it is still not clear, what is the "numerical optimization" in the title? Do the authors mean zero-order optimization?

The Abstract must bee informative and contaib ALL the essential information about the results of the paper so that a reader can understand the value of the peper without reading the whole paper. From the Abstract, it is not clear, for which kind of optimization problems the algorithm is designed: continuous/discrete, constrained/unconstrained, convex/con-convex, global/local optimization, large-scale/not large scale. In the end of the abstract, the authors mention constrained problems and engineering design problems. Such terms as "high soar", "verical stoop", "glide..." must be explained in terms op optimization theory and optimization methods in the Abstract.